# Conceptual Design and Multi-Disciplinary Computational Investigations of Multirotor Unmanned Aerial Vehicle for Environmental Applications

**Vijayanandh Raja** [1], **Senthil Kumar Solaiappan** [1], **Parvathy Rajendran** [2], **Senthil Kumar Madasamy** [1] and **Sunghun Jung** [3,*]

1   Department of Aeronautical Engineering, Kumaraguru College of Technology, Coimbatore 641049, India; vijayanandh.raja@gmail.com (V.R.); senthil.avionics@gmail.com (S.K.S.); senthilkumar.m.aeu@kct.ac.in (S.K.M.)
2   School of Aerospace Engineering, Engineering Campus, Universiti Sains Malaysia, Nibong Tebal 14300, Malaysia; aeparvathy@usm.my
3   Faculty of Smart Mobile Convergence System, Chosun University, Dong-gu, Gwangju 61452, Korea
*   Correspondence: jungx148@chosun.ac.kr

**Abstract:** This study focuses on establishing a conceptual design for a multirotor unmanned aerial vehicle (UAV). The objectives of this octocopter are to reduce the number of flight cancelations and car accidents owing to low-visibility issues and to improve abnormal environmental conditions due to the presence of smoke. The proposed octocopter contains a convergent–divergent [CD] duct-based storage tank, which provides a platform to store saltwater and allows it to fly in foggy zones. Fine saltwater is sprayed from the octocopter and dispersed into the low clouds, thereby altering the vapor's microphysical processes to break it up and improve visibility. The nature of the seawater and its enhanced fluid properties, due to the involvement of octocopter, creates the fluid flow mixing between atmospheric fluids and spraying particles, which increases the settling of foggy and smokey content groundward. For deployment, the conceptual design of the octocopter was initially constructed through analytical approaches. Additionally, three unique historical relationships were created. The standard engineering approaches involved in this work were stability analysis through MATLAB and fluid-property analysis through computational fluid dynamics (CFD) cum multiple reference frame (MRF) tools. The systematic model of this octocopter was developed by CATIA, and thereafter CFD and fluid–structure-interaction (FSI) analyses were computed, in ANSYS Workbench, on the octocopter for various environmental conditions. The aerodynamic forces on the drone, the enhancement of dynamic pressure by the presence of high amounts of rotors and nozzle sprayer, suitable material to resist aerodynamic loadings, and tests on the efficiency of the controller and its electronic components were investigated in detail. Finally, the proposed octocopter-based dynamic system was conceptually constructed.

**Keywords:** CFD; environmental drone; FSI; ocean water; stability analysis; UAV

## 1. Introduction

Many engineering applications have been fully implemented and used in the day-to-day activities of people. In view of the vast range of implementations, there is a perceived need for automatic systems to make life-related engineering applications more convenient, such as those regarding medical transportation, house cleaning, interior painting, and industry monitoring. In this context, unmanned aerial vehicles (UAVs) have been implemented in various applications. UAVs are a type of aircraft that can fly and execute their intended missions without the help of an onboard pilot.

The control of a UAV is very complicated and is usually executed based on one of two methodologies. The first involves a fully autonomous flight-control system. The

maneuvering of the UAV, its payload control, and its mission execution are handled entirely using preloaded programs. The second is a partially autonomous flight-control system. A ground-based pilot manages the primary controls using a remote controller, and a preloaded program manages secondary controls, such as altitude and attitude holding.

UAVs can be classified into several categories. Some of the primary classifications, such as those based on range, endurance, weight, or maneuverability, are fundamentally supported by the selection of the UAV. In UAV construction, the intended engineering application is a key factor; moreover, such decision needs to be finalized to select an appropriate UAV for further processing.

Three fundamental types of UAVs are most commonly implemented in engineering applications to efficiently execute missions: fixed-wing UAVs, rotary-wing UAVs, and multirotor UAVs, which were implemented in this study. The results showed that the octocopter is the preferred platform in multirotor UAV configurations, owing to its stability. Apart from the stability, the rotodynamic effect of the rotor plays a major role in the applications under examination, so a UAV with a high number of rotors was chosen for this investigation.

The target application of this work is to provide a clear visibility zone in environmentally critical transport terminals such as airports, railway stations, and bus stands. These terminals can be affected by foggy weather conditions, smoke generated from natural disasters, and other visibility-affecting issues. Poor visibility can affect the routine operations of these terminals. Collisions, such as unavoidable accidents between airplanes during takeoff and landing, or accidents between vehicles and humans, can create major problems. Additionally, minor problems can occur due to poor visibility, such as delays in departure and arrival of the targeted vehicles, and loss or delay in the communication systems. Thus, these major and minor problems need to be solved to ensure the smooth and successful operation of these terminals. In this regard, a dynamic mode-based unmanned aircraft system (an octocopter) is here proposed to troubleshoot the aforementioned critical environments. To reduce the poor visibility generated from unwanted environmental conditions such as fog and smoke, a highly dense seawater-relayed sprayer mechanism is proposed in this work. To execute this operation, a separate module is attached to the octocopter. The fundamental details of this module are included in the theoretical construction of the octocopter. The unique module comprises a storage tank, nozzle sprayer, and supportive joints, as shown in Figure 1. The first novelty involved in this work is the shape of the storage tank, which is a convergent–divergent duct-based construction. Aerodynamically, the convergent duct provides an increase in velocity, the divergent duct provides a decrease in velocity, and a combination of convergent and divergent ducts provides a high dynamic pressure for the fluid (seawater); thus, the proposed design can provide increased velocity of the seawater at the exit point of the tank. Additionally, the sprayer nozzle is installed at the tank's exit point, so the possibility of a further increase in the seawater velocity is quite high. In addition to these two factors, the rotodynamic effect of eight propellers can increase the induction of the velocity of the foggy/smoky fluid outside the UAV. Thus, highly energized seawater from the octocopter's tank can easily modify the properties of foggy/smoky fluids, thereby pushing them to settle to the ground (clearly shown in Figure 2). The visibility in and around terminals will therefore increase because of the settling of the foggy/smoky fluids. Since the proposed idea is UAV-based dynamic equipment, the area coverage of the seawater sprayer around the terminals is higher than that of existing cleaning systems.

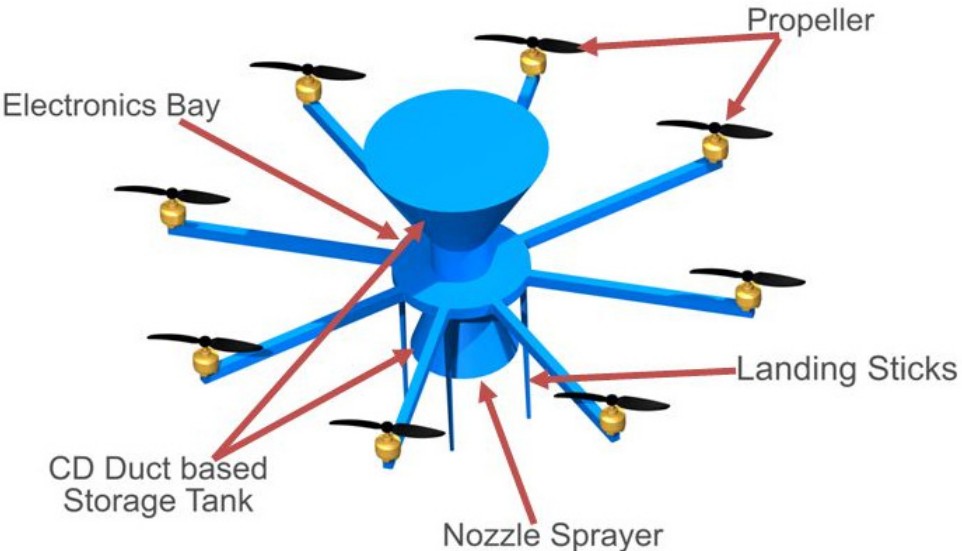

**Figure 1.** Conceptual design of proposed octocopter, with its components.

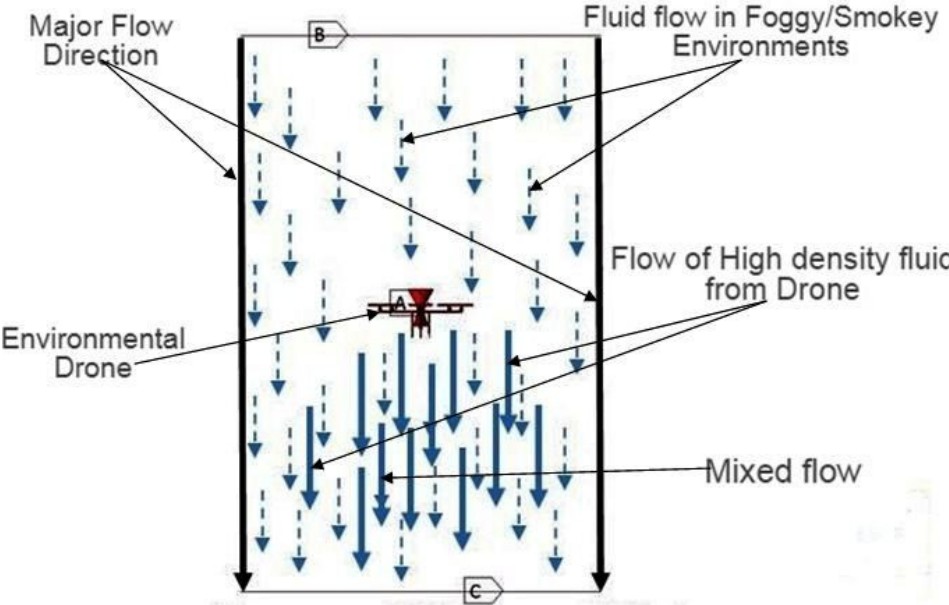

**Figure 2.** Concept involved in this study.

## 2. Literature Survey

Yeong et al. [1] investigated the propeller of a UAV. Computational fluid dynamics (CFD) and experimental approaches were used to select the design parameters for a two-bladed UAV propeller. Standard theoretical calculations were used to estimate the boundary conditions of the CFD problem. The principal estimated conditions were the thrust generated by the propeller, velocity conditions of the propeller, and Reynolds' number of the flow. The same procedure was planned as an extension to the current study, although the working propellers were different.

Ong et al. [2] theoretically constructed high-payload-based multirotor UAVs for complicated applications. The authors derived the relationship between the payload weight and the overall weight of the UAV using historical data. The theoretical estimations were extended to the selection of the UAV's primary components. Finally, a coaxial-propeller-based Tricopter was constructed and tested in complicated environments.

The current study considered successful surveillance for high-payload and complex applications; therefore, theoretical derivatives were implemented based on a literature

survey [3]. The current study's targeted application is the clearance of dirty/dusty environmental conditions through non-hazardable liquids, such as seawater and urea. Thus, the liquid weight was fixed at 1000 g, and the subordinate equipment was finalized. The cumulative payload weight of the multirotor UAV was determined in the context of the proposed relationship.

Many researchers have modeled and controlled multirotor vehicles. In [4], the mathematical modeling of an octocopter was developed by including the thrust and torques generated by the propellers and adding gyroscopic torque. A full-state feedback-based hybrid controller scheme was designed against a linearized model based on motor dynamics.

In [5], a physical model of an octocopter was developed using the theory of rigid-body mechanics and aerodynamics. The unknown parameters in this model were determined through several identification experiments. An attitude-estimation algorithm was designed and implemented on the target hardware. The algorithm is referred to as a nonlinear complementary filter, and it uses a quaternion rotation representation and onboard measurements to compute an estimate of the current aircraft attitude.

Two different attitude controllers have been designed and evaluated [5]. The first controller was based on proportional–integral–derivative techniques, which are commonly used in multirotor flight-stabilization systems. The second controller used a novel control structure based on the L1 adaptive-control techniques. A baseline attitude proportional-derivative (PD) controller was augmented with an L1 adaptive controller in the rate-feedback loop. The two controller structures were compared using a simulation environment based on the developed octocopter model.

In [6], the theoretical and analytical portions of the octocopter modeling and control were considered, focusing on the development process. A complete nonlinear model of the octocopter was derived, using Newton's second law of motion in a rotating reference frame. Additional effects, such as aerodynamics, precession torque, and motor dynamics, were analyzed and modeled.

In this study, control laws and approaches for controlling the vehicle in six degrees of freedom were proposed for position and attitude control. The position controller uses a nonlinear Lyapunov-based controller, where the control inputs are converted to the desired attitude-reference inputs and sent to the attitude controller. The attitude-control method was based on a full-state feedback controller, and a reduced observer was used to observe motor dynamics. The control laws were evaluated using a Simulink model in which a complete nonlinear system was implemented.

In [7], computational and experimental investigations were carried out on an octorotor-based small multirotor UAV. Comprehensive performance analyses were computed and tested based on the predominant consideration of the rotor spacing and its variations. The fundamental object of this study was a medium-sized propeller, with dimensions of 15.75 in diameter and six-inch pitch. The rotodynamic effect of the propeller can create a cascade effect, thereby affecting the entire performance of the octocopter; therefore, this study provides significant information about perfectly positioning the eight rotors in the octorotor system. The computational investigations were computed using an octorotor with the discretization category of an unstructured grid formation. Through the inclusion of the study [7], the design ratio between the diameter of the propeller and the length of the connecting arm was fixed at two for the current design. Through this high design ratio, the possible cascade-based effect on performance should never occur in the proposed design.

In [8], the CFD and FSI computational approaches were imposed on an unmanned amphibious vehicle [UAmV], in which the aerodynamic-dependent initial and boundary conditions were useful data for this study. In particular, two important observations noted from the study [8] are that the k-epsilon turbulence model was imposed for the prediction of wake in and around the UAmV, and that the [semi-implicit method for pressure-linked equations] SIMPLE-based pressure- and velocity-coupling approach was implemented for the internal computations. In addition to these conditions, the design procedure involved

in the study for four-bladed amphibian rotors has been adopted for the design of the current propulsive system.

In [9], the authors computationally investigated the performance of various ducts and ducted propellers using a CFD approach. The ducts involved in this comparative investigation were convergent, divergent, and convergent–divergent. Based on the high induced-velocity factor, the convergent–divergent duct was selected as the best performer over the other two ducts. An unstructured discretized control volume was constructed, in which the k-epsilon turbulence model was imposed to predict the flow inside the ducts. In this study, the implementation of a CD duct-based storage tank in an octocopter emerged because of the need for a high fluid velocity at the exit of the tank.

In [10], the authors computed fluid dynamic behavioral analysis for a hexacopter using ANSYS Fluent. The major focus was on the variations in the angle arrangement of the connecting arms in the frame of the UAV. Owing to the angle variations of the connecting arms in the UAV, the aerodynamic forces were moderated and twisted. Because of this segregation of aerodynamic forces, additional sideward forces were induced, which provided static stability to the multirotor UAV. The assembly of connecting arms for the current study was constructed using a literature survey [10].

The main observation of the study [11] was the computational procedure for the execution of the moving reference frame (MRF) approach on the propeller. In the current work, the MRF analyses are planned to be imposed on eight different rotors, so the implementation of MRF is unavoidable. From the study [11], it was also found that suitable initial and boundary conditions are also suitable for transient-flow computations. Through these inputs, the convergence of the current computation occurs without any computational oscillations.

The objective of the current study is to develop an octorotor-configured multirotor UAV for clear visibility-based environmental applications in terminals such as airports, railway stations, and bus stands. The focused engineering approaches planned to investigate the deployment behavior in and around the octorotor are: computational analyses, wherein computational fluid dynamics (CFD) is focused on investigating aerodynamic drag over an octocopter and deployment fluid behavior of various flows; fluid-structure-interaction analysis, focused on choosing a suitable material to effectively resist the targeted environments of deployment; and control-dynamic analysis, used to study the working performance of the controller and its linked electronics components. The novelty involved in this work is the advanced convergent–divergent duct-based storage tank, unique analytical approach-based design construction of the propeller, and dynamic platform [octocopter] to solve visibility-relevant issues at various terminals. Details of the concept of this multidisciplinary investigation are shown in Figure 2.

## 3. Component Selection and Criteria

In general, the critical factor in the aeronautical field is the weight, as it plays an influential gyroscopic role in the construction of all components. All primary systems, such as propulsive, aerodynamic, and flight-control systems, depend on the aircraft weight. Hence, the component selection of an aircraft must be considered from the perspective of weight. Similarly, this study also determined the components of the octocopter by relying on weight factors. Initially, a standard trial-and-error approach was implemented, in which fixed-mass procedures were followed. In a fixed-mass methodology, two types of masses play a vital role in UAV component selection: the overall mass of the UAV and the payload mass. The payload weight-based approach was chosen in this work, owing to the clear application plan for the UAV. In this targeted application, the essential supporting parameters (such as the storage tank capacity, speed of the fluid, design of the nozzle sprayer, center of gravity variations, endurance of the spray process, and recycling process) are known, and guided the determination of the payload weight. Generally, the payload

weight of the UAV is comprised of secondary and primary categories, as expressed in Equation (1).

$$W_{Pl} = W_{Primary\ Payload} + W_{Secondary\ Payload} \tag{1}$$

In this work, the storage capacity of the sea water container is assumed to have a weight of 1000 g. In the secondary payload category, the nozzle sprayer and its holding devices contribute significantly. The weight of the nozzle sprayer was estimated to be 50 g. A glass fiber-reinforced polymer (GFRP)-based composite material is suggested for the construction of a C–D duct-based storage tank so that the approximate weight of the tank is 200 g. Thus, the payload weight is estimated as 1.25 kg.

### 3.1. Theoretical Calculation

In this study, standard literature surveys and historical data were consulted, and the initial conditions were determined [12]. Using these related conditions, a UAV was constructed theoretically. Equation (2) was obtained from historical data collected from various previously completed octocopters. The comprehensive relationship between the payload and the overall weights of various octocopters is shown in Figure 3.

$$W_{Pl} = 0.30\ W_O \tag{2}$$

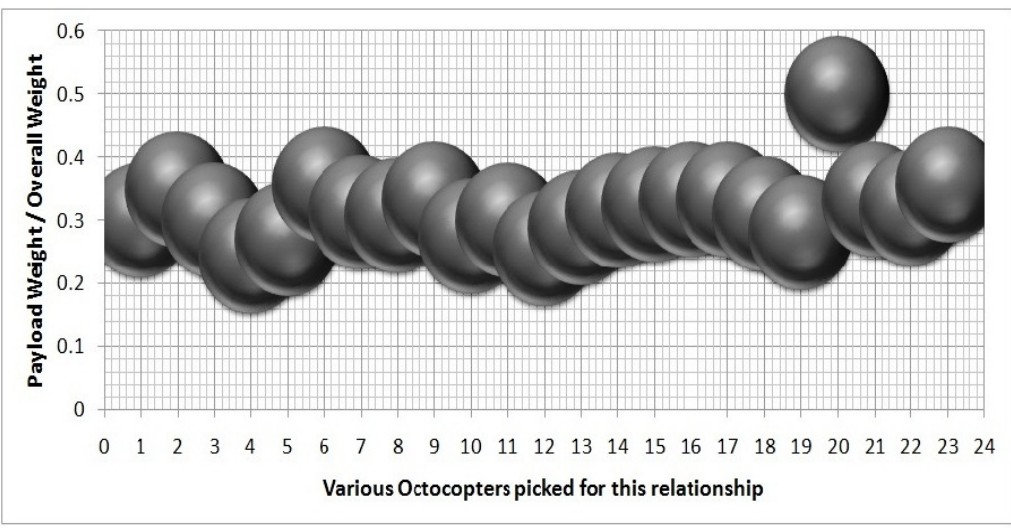

**Figure 3.** Historical relationship between payload weight and overall weight of octocopter.

The weights of the storage device, sprayer, other accessories, and payload were 1000 g, 50 g, 200 g, and 1250 g, respectively. Therefore, the overall weight can be calculated as $\frac{1250}{0.30} = 4166.67$ g.

Hence, the minimum thrust/lift required by a single propeller of the octocopter was equal to $\frac{4166.67}{8} = 520.83$ g at a normal level. At the maximum level, the thrust-to-weight ratio is assumed to be 1.75, and at a normal level, the ratio is equal to one. With these significant inputs, the primary components were estimated, in which the propeller, motor, battery, and electronic speed controller (ESC) were important. The propeller for this octocopter was designed using the conventional formula in Equation (3). The corresponding calculations were as follows:

Thrust requirement of the $\sin$ gle propeller at maximum level $= \frac{1.75*4166.67}{8} = 911.5$; and

$$T = 0.5 \times \rho \times A \times \left[ (V_{UAV})^2 - (V_0)^2 \right]. \tag{3}$$

From the field work, the gust aerodynamic velocity [$V_0$] was estimated to be 10 m/s, and the maximum speed of the UAV [$V_{UAV}$] was assumed to be 30 m/s. Accordingly, the dimensions were calculated as follows:

$$911.5 \times 9.81 = 0.5 \times 1.2256 \times A \times \left[(30)^2 - (10)^2\right];$$
$$A = \frac{911.5 * 9.81}{490.24} = \frac{8.933}{490.3} = 0.01822;$$
$$\text{and thus the diameter} = 6 \text{ in.}$$

After the estimation of the diameter, the pitch estimation plays a major role in the construction of the propeller. In this regard, the historical relationship between the pitch and diameter of relevant propellers was examined and thereby showed the relationship suitable for a six-inch-diameter propeller, seen in Equation (4). Figure 4 shows the variation in the aforementioned design parameters.

$$\frac{\text{Pitch}}{\text{Diameter}} = 0.75 \Rightarrow \text{Pitch} = 0.75 * \text{Diameter} \Rightarrow 0.75 * 6 = 4.5 \text{ inches} \tag{4}$$

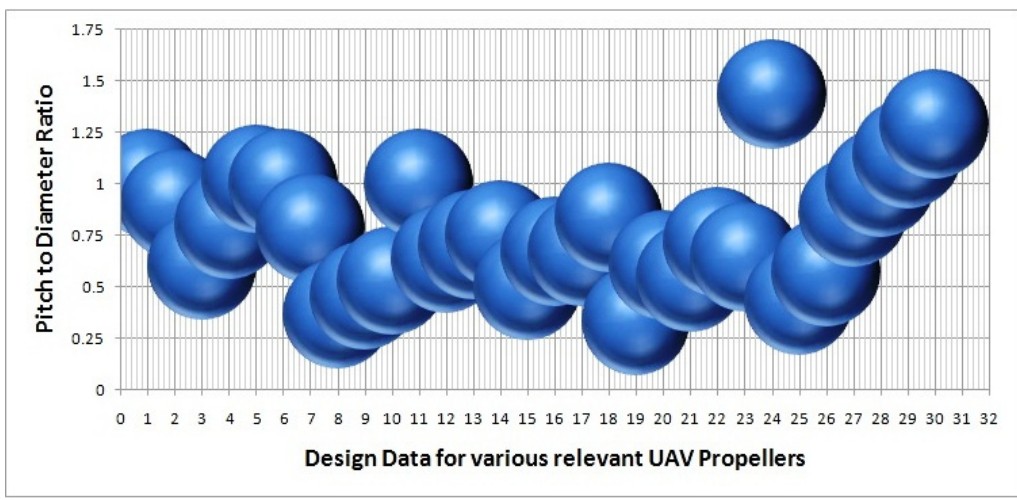

**Figure 4.** Historical relationship between pitch versus diameter of the octocopter's propellers.

Thus, a 6 × 4.5 propeller was estimated for this application, and the total weight of the entire propeller setup was determined to be 120 g (8 × 15 g).

### 3.2. Frame Design and Weight Estimation

From the literature survey [1–17], it was found that the octocopter-based frame has been used for high-stability-based/high-payload-based applications; thus, it was implemented in this work. The application addressed in this work was very challenging; therefore, instead of a complicated design, a simple configuration was imposed, as follows:

$$\text{Angle of the Connecting arm} = \frac{\text{Overall rotational degree}}{\text{Arm count of UAV frame}} = \frac{360}{8} = 45^\circ.$$

With the help of the cosine theorem, the length of the arm was obtained using Equation (5), which assumes that all the arms are of the same length, as follows:

$$\text{Length of the arm} = \sqrt{\frac{(\text{Diameter of the Propeller})^2}{2 \times (1 - \cos \theta)}}$$
$$\text{Length of the arm} = \sqrt{\frac{(6)^2}{2 \times (1 - \cos(45))}} \Rightarrow 7.84 \text{ inches} = 0.199136 \text{ m} \approx 0.20 \text{ m.} \tag{5}$$

In multirotor UAVs, the fineness ratio (FR) plays a significant role in the design stage. The conventional relationship of the FR ratio is shown in Equation (6), which plays a predominant role in the design of the length and thickness of the connecting arms, as follows:

$$\text{FR} = \frac{\text{Length of the connecting arm}}{\text{Thickness of the connecting arm}} \tag{6}$$

Generally, an FR ratio of 8-to-10 is more suitable for long arms, so the FR ratio was assumed to be 10, as follows:

$$10 = \frac{\text{Length of the arm}}{\text{Thickness of the arm}} \Rightarrow \text{Thickness of the arm} = \frac{0.2}{10} = 0.02 \text{ m.}$$

In general, to provide a better platform for other components and easy attachments, a square cross-section provides a nearly perfect fit for the connecting arms; thus, the same cross-sectional shape was used in this work. Therefore, the dimensions of a single connecting arm were $0.2 \times 0.02 \times 0.02 = 0.00008 \text{ m}^3$. A previous study [8] has shown that a GFRP composite can resist aerodynamic loads; thus, the same material was chosen as the construction material for the octocopter frame. A structural simulation was performed using a GFRP with an average density of approximately $1500 \text{ kg/m}^3$. Thus, the single-arm weight was 120 g, and the overall weight of the connecting arms was 960 g [11–15].

### 3.3. Landing Gear Design and Weight Estimation

The landing gear is an important part of a multirotor UAV as it needs to carry the entire weight of the unmanned vehicle. Therefore, the design of the landing gear of the UAV focuses primarily on the number of sticks, length of the sticks, cross sections of the sticks, and position of the landing sticks. In this study, the center of gravity of the UAV lies in the center of the design and must compensate for the storage tank attachment. Four landing sticks were used in the conceptual design.

As mentioned earlier, the bottom divergent portion of the proposed CD-design -based duct needs to be enlarged in order to do further increment in the exit velocity of storage fluid at exit place of tank. Thus, the length of the landing sticks should be fixed at a higher value to handle complicated atmospheric conditions. So, value of two is better for the ratio of the arms to the landing stick in complicated application-based UAVs, so the stick ratio (SR) was assumed to be two, as given in Equation (7):

$$\text{Stick Ratio} = \frac{\text{Length of the Landing stick}}{\text{Length of the arm}} = 2; \text{ thus,}$$

$$\text{Length of the Landing stick} = 2*\text{Length of the arm} = 2*0.2 = 0.4 \text{ m.} \tag{7}$$

The same ratio was extended to select the cross-section of the landing sticks, so the surface area of the square landing sticks was $0.04 \times 0.04$. Therefore, the overall weight of the landing stick was 960 g. To construct the connecting arms and landing sticks, a 1:10 scale ratio was used for the major dimensions.

### 3.4. Estimation of Thrust and Specifications

In this work, the thrust-to-weight ratio was assumed to be two for aggressive gust-load conditions. The complete relationship is given in Equation (8) as follows:

$$\text{Thrust to Weight Ratio} = \frac{\text{Thrust}*\text{Number of Rotor}}{\text{Weight of the UAV}}$$

$$\text{Thrust requirement of the single propeller} = \frac{2 \times 4426.8}{8} \Rightarrow 1106.7 \text{ g.} \tag{8}$$

From Equation (3), the calculations are as follows:

$$10.856727 = 0.5 \times 1.2256 \times \pi \times r^2 \left[ (V_{\text{UAV}})^2 - (4)^2 \right].$$

where "r" is the radius of the propeller (0.0719655 m), and $V_{UAV}$ is the forward velocity of the UAV in m/s, which is also equal to the maximum induced velocity provided by the propellers. The corresponding calculations are as follows:

$$10.856727 = 0.009965454 \left[ (V_{UAV})^2 - (4)^2 \right];$$

$$0.0099655 \left[ (V_{UAV})^2 \right] = 10.857 + 0.15945 \Rightarrow 0.0099655 \left[ (V_{UAV})^2 \right] = 11.0162; \text{ and}$$

$$(V_{UAV})^2 = 1105.4363 \Rightarrow (V_{UAV}) [\text{induced speed of the UAV owing to heavy gust load}] = 33.25 \text{ m /s}.$$

The angular-to-linear velocity formula was estimated using Equation (9) as follows:

$$V_{UAV} = r * \omega . \tag{9}$$

Here, $V_{UAV}$ is the velocity of the octocopter in m/s, "r" is the radius in m, and "$\omega$" is the angular velocity in rad/s. The RPM to linear velocity was determined using Equation (10), as follows:

$$RPM = \frac{V_{UAV}}{0.10472 \times 0.0719655}$$
$$RPM = \frac{V_{UAV}}{0.00753622716} \Rightarrow \frac{33.24810169808709}{0.0075362271} = 4412. \tag{10}$$

For a heavy-gust load, the maximum RPM of the single propeller was estimated to be 4412; however, in most cases, the UAV needs to maintain only an average rotational speed. Thus, a 2206 KV rate-based motor was the best option for this UAV. The total weight was 180 g (8 × 22.5 g). The best battery finalized for this operation was 10Ah in capacity, 4-S connected, and 35C in discharge rate, and the overall weight was approximately 500 g. Generally, a 30-A current draw rate-based ESC is suitable for handling 4S batteries, so the same ESCs were short-listed for this operation, and the total weight of the ESC with wires was 72.8 g (8 × 9.1 g). Other electronics such as flight-control boards, receivers, and power-distribution boards were selected based on the aforementioned components, totaling approximately 100 g [7–10]. The total UAV weight = [1250 + 120 + 180 + 500 + 196 + 72.8 + 960 + 960] = 4238.8 g. The calculated weight of the entire octocopter was close to the historically obtained weights; thus, with these components, the conceptual design phase of this UAV was initiated.

## 4. Conceptual Design of the Octocopter

For the construction of the conceptual design, the pivotal points included the ideation of the conceptual design concerning the application, requirements of the components, estimation of weight, selection of main parameters, selection of sub-parameters, conceptual design layout and configuration, amalgamation of performances, and optimization characteristics.

### 4.1. Outline of Conceptual Design

In general, an octocopter is a flying craft with eight functioning propellers powered by eight motors. Octocopters are more capable, reliable, and stable than hexa and quadcopters. Owing to its robust functional parameters, an octocopter can play a significant role in operational applications. Therefore, the implementation of a vacuum cleaner can provide an example.

A vacuum cleaner works based on the flow of air from a high-pressure region to a low-pressure region. When an electric motor spins at a high velocity coupled with a fan, it creates an area of low pressure inside the suction hose. Thus, the particles and debris in the air are sucked into the suction hose owing to the pressure difference. A high-efficiency particulate absorber (HEPA) filter can be used to sanitize and remove allergens from collecting bags. In our conceptual design, a centrifugal motor (typically a universal motor) was mounted on the hub's center of the octocopter. When the motor and fan were rotated, it sucked in the atmosphere with a convergent inlet passage.

The particles entered the inlet via an HEPA filter fitted with a collecting bag (a tiny bag or box). This filter minimized the particles that could disturb the motor. A HEPA filter can remove 99.97% (ASME standard) of particles and debris greater than or equal to 0.3 μm. When the sucked air particles flowed behind the motor with a divergent passage to the HEPA filter mounted with the collecting bag, the depressurized and particle-free air entered the atmosphere.

### 4.2. The Octocopter

The typical views of the conceptual design of the octocopter are shown in Figures 5 and 6, in which the predominant tool used for this construction was CATIA. Appropriate selection of parameters provided a potential profile for the design of the octocopter. The design, coupled with the centrifugal motor in the convergent inlet passage, sucked out particles via a narrowing path.

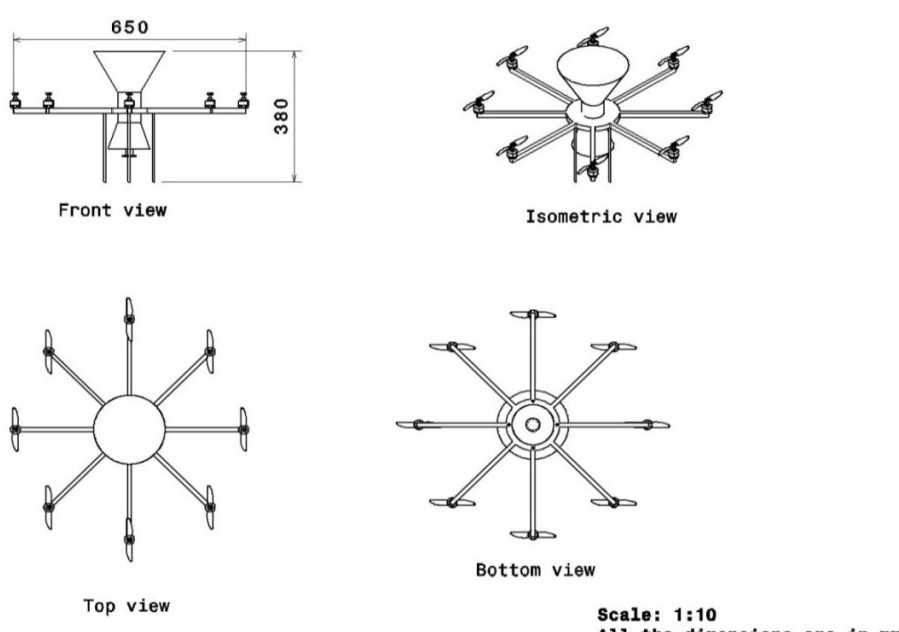

**Figure 5.** Conceptual design of an octocopter for environmental applications.

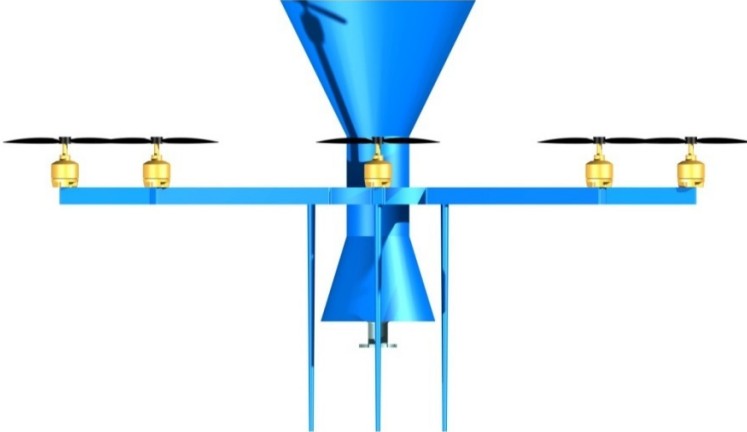

**Figure 6.** Typical front view of conceptual design of an octocopter.

The collected debris from the HEPA filter was dumped into the collecting bag, which was mounted below the hub and surrounded by the motor outlet, and the flow was again treated with a filter. Figure 6 shows the front view of the conceptual octocopter fitted with a retractable landing gear for appropriate applications. The propeller diameter was

143.93 mm, distance between the propellers was 645.65 mm, and the distance from the propeller to the central hub was 322.83 mm.

### 4.3. Estimation of $C_L$, $C_D$, and Moment of Inertia

4.3.1. Vertical Take-Off

The aerodynamic force equilibrium of this UAV for vertical takeoff and landing was defined using Equation (11) as follows:

$$\text{Lift} = \text{Weight} + \text{Drag} \Rightarrow \text{Lift} - \text{Weight} - \text{Drag} = 0 \tag{11}$$

In the multirotor-based propulsive system, the performance factors such as coefficient of lift and coefficient of drag to be estimated as follow as:

$$\frac{1}{2}\rho(V_{UAV})^2 S_P C_L = \text{Lift} = \text{Thrust} = T = 0.5 \times \rho \times A_P \times [(V_{UAV})^2 - (V_0)^2];$$

$$C_L = \frac{0.5 \times \rho \times A_P \times [(V_{UAV})^2 - (V_0)^2]}{\frac{1}{2} \times \rho \times S_P} \Rightarrow C_L = 1 - \frac{V_o^2}{(V_{UAV})^2}; \text{ and}$$

$$C_L = 1 - \frac{4^2}{(33.248)^2} \Rightarrow C_L = 0.98552.$$

From Equation (11), relatzionships can be derived as follows:

$$\left(\frac{1}{2}\right) \times \rho \times (V_{UAV})^2 \times S_P \times C_L = (4.43 \times 9.81) + \left(\frac{1}{2}\right) \times \rho \times (V_{UAV})^2 \times S_D \times C_D;$$

$$0.5 \times 1.2256 \times 33.25 \times 33.25 \times 8 \times 0.0163 \times 0.98552 = (4.43 \times 9.81) + \left(\frac{1}{2}\right) \times \rho \times (V_{UAV})^2 \times S_D \times C_D;$$

$$86.85 - 43.426908 = \left(\frac{1}{2}\right) \times \rho \times (V_{UAV})^2 \times S_D \times C_D; \text{ and}$$

$$C_D = \frac{70.8601}{(V_{UAV})^2 S_D} \Rightarrow C_D = \frac{0.06411}{[8*0.0163 + 0.032 + 0.0314]} \Rightarrow C_D = 0.3314.$$

where, "$C_L$" is coefficient of Lift, "$C_D$" is coefficient of Drag, "$S_P$" is planform area of Propeller, "$A_P$" is cross sectional area of the propeller, "$S_D$" is drag contributing outer area of the octocopter, $V_{UAV}$ is the induced velocity by the UAV's propeller in m/s, and $V_0$ is the atmospheric fluid velocity in m/s.

4.3.2. Forward Maneuvering

From the conventional free-body force inputs of multirotor UAV, for the forward speed, it can be determined that the angle $\alpha$ between the horizontal plane and the reference plane of the UAV arm is essential for estimating the forward speed and its subordinate specifications. The calculations are as follows:

$$\begin{aligned} L \times \cos(90 - \alpha) - \text{Drag} = 0; \\ L \times \cos(90 - \alpha) = \text{Drag}; \end{aligned} \tag{12}$$

$$\begin{aligned} L \times \sin(90 - \alpha) - \text{Weight} = 0; \text{ and} \\ L \times \sin(90 - \alpha) = \text{Weight}. \end{aligned} \tag{13}$$

Four different angles (15°, 30°, 45°, and 60°) underwent aerodynamic performance estimations; the performance was better at 30° than at other angles. Therefore, the same angle was used in this work for the estimation of aerodynamic performance and stability analysis , $\alpha = 30°$, as follows:

$$L \times \sin(30) = \text{Weight}; \tag{14}$$

$$L \times \cos(30) = \text{Drag}; \tag{15}$$

$$\text{Upward Force } (L) = \frac{\text{Weight of the octocopter}}{\sin(30)}; \tag{16}$$

$$\text{Forward Force (T)} = \frac{\text{Drag over the octocopter}}{\cos(30)} \; ;$$
$$\text{Upward Force (L)} = \frac{4.4268}{0.5} = 8.8536 \text{ kg; and} \qquad (17)$$
$$8.8536 \times \cos(30) = \text{Drag} \Rightarrow 8.8536 \times 0.866 = \text{Drag} = 7.6672176 \text{ kg .}$$

Newton's Second Law is as follows:

$$F = m \times a \qquad (18)$$

Here, "F" is the resultant force in Newtons, "m" is the mass of the UAV in kg, and "a" is the acceleration of the UAV in $m/s^2$.

$$F = 8.8536 - 7.6672176 = 1.1863824 * 9.81 = 11.638411344 \text{ N.}$$

Thus,

$$a = \frac{11.638411344}{7.6672176} = 1.518 \Rightarrow \int a = v \Rightarrow \int_0^t 1.518.dt \Rightarrow v = 1.518 \times t.$$

In the above, "t" is the time taken at the phase, assumed to be 20 s; therefore, the velocity is calculated as follows:

$$V \text{ (forward speed)} = 1.518 * 20 = 30.36 \text{ m/s.}$$

From Equation (14), we have calculations as follows:

$$\tfrac{1}{2}\rho v^2 SC_L \times \sin(30) = 4.4268 \times 9.81;$$
$$[0.5 \times 1.2256 \times (30.36)^2 \times 8 \times \pi \times (0.0719655)^2]C_L \times 0.5 = 4.4268 \times 9.81;$$
$$C_L = \frac{4.4268 \times 9.81}{0.5 \times 1.2256 \times 30.36 \times 30.36 \times 8 \times 3.14 \times 0.0719655 \times 0.0719655}; \text{ and}$$
$$C_L = \frac{43.426908}{73.484} = 0.5911.$$

From Equation (15), we have calculations as follows:

$$L \times 0.866 = D = 7.6672176 \text{ kg} = 7.6672176 \times 9.81 = 75.22 \text{ N;}$$
$$\tfrac{1}{2}\rho v^2 SC_D = 75.22; \text{ and}$$
$$C_D = \frac{75.215404656}{0.5 \times 1.2256 \times 30.36^2 \times 0.1935} \Rightarrow \frac{75.22}{109.3} \Rightarrow C_D = 0.6882.$$

### 4.3.3. Estimation of Moment of Inertia

The moment of inertia is one of the primary factors contributing to the stability of a UAV and its effects. The moments of inertia of the motor, propeller, and entire octocopter are crucial for the stability calculations for all maneuvers. In this regard, the required moments of inertias were evaluated [3] as follows: the moment of inertia of the octocopter was $I_{oxG} = 0.079$ kgm$^2$, $I_{oyG} = 0.084$ kgm$^2$, $I_{ozG} = 0.109$ kgm$^2$, the moment of inertia of the propeller was $I_{oxG} = 0.000002493$ kgm$^2$, $I_{oyG} = 0.00007081$ kgm$^2$, and $I_{ozG} = 0.00007103$ kgm$^2$.

## 5. Mathematical Modeling and the Control of Attitude Dynamics

This section discusses the design of the attitude controllers for the octocopter. For this purpose, a linearized model of attitude dynamics is presented, rather than full-vehicle dynamics [12,13]. To create the dynamic model of the octocopter, a body coordinate frame, as illustrated in Figure 7, was used. All forces and torques generated on the octocopter platform can be expressed in the body frame, and are defined as follows:

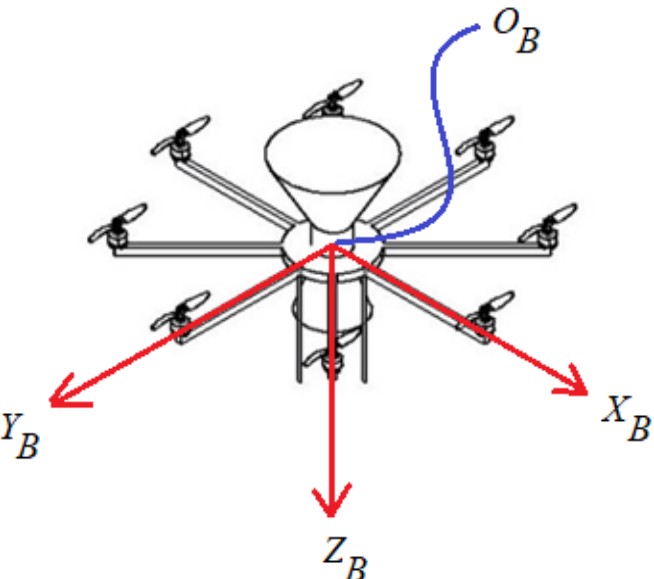

**Figure 7.** Illustration of body frame.

The origin ($O_B$) of the frame was placed at the center of mass of the octocopter.

○ The X-axis ($X_B$) is pointing forward along the structure.
○ The Y-axis ($Y_B$) points along the structure to the right.
○ The Z-axis ($Z_B$) points downward to complete a right-handed system.

### 5.1. Linearized Model of Attitude Dynamics

The linearized model for the angular velocities and attitude of the octocopter was obtained by keeping the vehicle in the hovering state and assuming decoupled attitude angles. To obtain the model, the forces and torques created by gyroscopic and gravity effects and aerodynamic effects owing to airflow were neglected. Only the input torques generated by the rotors alone was considered [5–7]. By considering the dynamics of the angular velocities and their rates of change, Equation (19) for the attitude dynamics was determined as follows [14–23]:

$$\begin{bmatrix} \dot{\phi} \\ \dot{\theta} \\ \dot{\Psi} \end{bmatrix} = \begin{bmatrix} 1 & \sin\phi\tan\theta & \cos\phi\tan\theta \\ 0 & \cos\phi & -\sin\phi \\ 0 & \sin\phi\sec\theta & \cos\phi\sec\theta \end{bmatrix} \begin{bmatrix} p \\ q \\ r \end{bmatrix}, \tag{19}$$

where "$\phi$" is the roll angle, "$\theta$" is the pitch angle, "$\Psi$" is the yaw angle, "$p$" is the body roll rate, "$q$" is the body pitch rate, and "$r$" is the yaw rate. However, in the hovering state, $\phi = \theta = 0$. Therefore, Equation (19) becomes,

$$\begin{bmatrix} \dot{\phi} \\ \dot{\theta} \\ \dot{\Psi} \end{bmatrix} = \begin{bmatrix} 1 & 0 & 0 \\ 0 & 1 & 0 \\ 0 & 0 & 1 \end{bmatrix} \begin{bmatrix} p \\ q \\ r \end{bmatrix}; \text{ and}$$

$$\begin{bmatrix} \dot{\phi} \\ \dot{\theta} \\ \dot{\Psi} \end{bmatrix} = \begin{bmatrix} p \\ q \\ r \end{bmatrix} = \begin{bmatrix} \omega_x \\ \omega_y \\ \omega_z \end{bmatrix}, \tag{20}$$

where $\omega_x$, $\omega_y$ and $\omega_z$ are the vehicle's angular velocities in the roll, pitch, and yaw axes, respectively. Moreover, the relation between the angular velocity "$\omega$" of the vehicle and the input torque "$\tau$" is given as follows:

$$\begin{bmatrix} \tau_x \\ \tau_y \\ \tau_z \end{bmatrix} = \begin{bmatrix} I_{xx} & 0 & 0 \\ 0 & I_{yy} & 0 \\ 0 & 0 & I_{zz} \end{bmatrix} \begin{bmatrix} \omega_x \\ \omega_y \\ \omega_z \end{bmatrix}, \tag{21}$$

where $\tau_x$, $\tau_y$, and $\tau_z$ are the input torques from the rotor for the roll, pitch, and yaw axes, respectively; and $I_{xx}$, $I_{yy}$, and $I_{zz}$ are the moments of inertia of the octocopter about the roll, pitch, and yaw axes, respectively.

From Equation (20), we can write:

$$\begin{bmatrix} \dot{\omega}_x \\ \dot{\omega}_y \\ \dot{\omega}_z \end{bmatrix} = \begin{bmatrix} \ddot{\phi} \\ \ddot{\theta} \\ \ddot{\Psi} \end{bmatrix}. \tag{22}$$

Equations (20)–(22) can be summarized as follows:

$$\left.\begin{array}{l} \dot{\phi} = p = \omega_x \\ \dot{\theta} = q = \omega_y \\ \dot{\omega} = r = \omega_z \\ \ddot{\phi} = \frac{\tau_x}{I_{xx}} \\ \ddot{\theta} = \frac{\tau_y}{I_{yy}} \\ \ddot{\Psi} = \frac{\tau_z}{I_{zz}} \end{array}\right\}. \tag{23}$$

### 5.2. Motor Dynamics

The complete control system used in the octocopter's attitude control needs to consider the dynamics of the motors. The octocopter uses eight brushless, DC motors. Usually, a first-order low-pass filter approximates the motor dynamics based on the experimentally obtained values. Here, the motor's mathematical model was obtained from the electrical and mechanical properties of the motor. The transfer-function model of the DC motor and the load (propeller) is shown in block diagram form in Figure 8 [22].

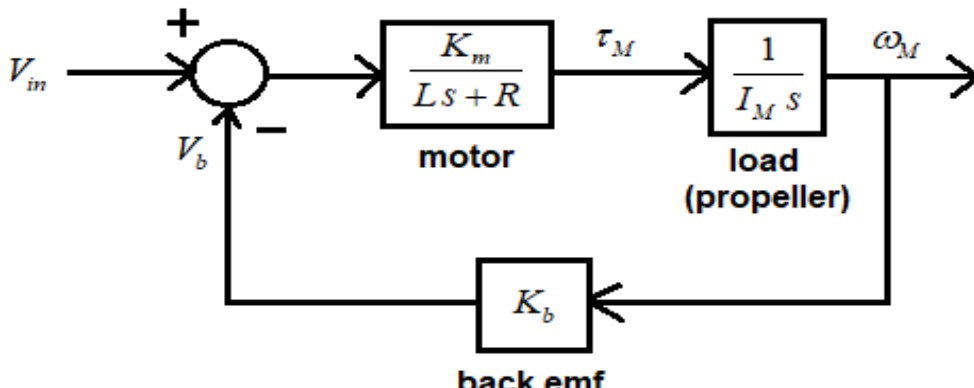

**Figure 8.** Block diagram of brushless, DC motor for a single phase with a load attached.

In Figure 8, $V_{in}$ represents the supplied phase voltage, $V_b$ represents the phase voltage back emf, $\tau_M$ represents the torque developed by the motor, $I_M$ represents the moment of inertia of the rotor and attached propeller, and $\omega_M$ represents the angular velocity of the propeller. The motors used in this study were of type 1000 KV and the specifications of the selected DC motors are listed in Table 1.

**Table 1.** 1000 KV Brushless DC motor specifications.

| Parameter | Value |
|---|---|
| Phase resistance, R | 0.128 Ohms |
| Phase inductance, L | 0.0000184 H |
| Phase inductance, L | 0.0000184 H |
| Torque constant, $K_m$ | 0.008 Nm/A |
| Back emf constant, $K_b$ | 0.00955 Volts/(rad/s) |
| Rotor inertia, I | |
| About *x*, *y*-axes | $8.5 \times 10^{-6}$ kg·m$^2$ |
| About *z*-axis | $4.964 \times 10^{-6}$ kg·m$^2$ |

### 5.3. Attitude-Control Loops

In general, the purpose of the attitude controller is to obtain a faster and more stable attitude response from the octocopter. The inputs to the attitude controller are the desired roll angle, pitch angle, and yaw rate, and the controller's output is the varying voltage for the motor to produce the required torque to obtain the desired output. The structure of a roll attitude control loop for a vehicle with torque from a single motor is shown in Figure 9. Nevertheless, in the simulation, simultaneous torques from the three motors was considered to obtain the unbalanced torques about the "*x*" and "*y*" axes to obtain the roll and pitch motions, respectively [14–17].

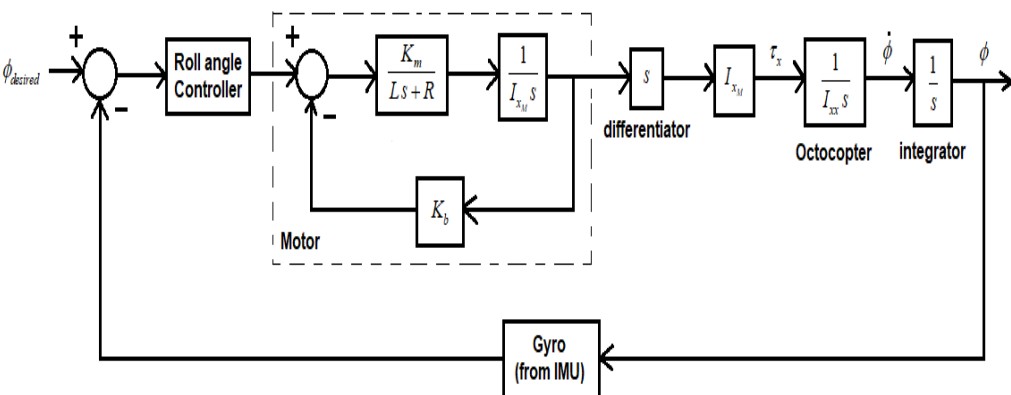

**Figure 9.** Roll attitude control loop.

The control loop structure of a pitch attitude is similar to that of the roll control loop, but the moment of inertia values change; the control loop is shown in Figure 10. For the yaw axis, the yaw rate should be controlled instead of the yaw angle. The octocopter can rotate freely in yaw, making it unnatural for a user to input the desired yaw angle [5]. Therefore, the yaw control loop structure is slightly different from the roll and pitch control loops, as shown in Figure 11.

### 5.4. Controller Design

The design of the attitude controller and its gains were based on the limitations in the root locus and difficulties in hardware implementations for higher gain values. Considering these constraints, the specifications for the roll and pitch attitude controllers were defined as shown in Table 2.

**Table 2.** Controller design specifications.

| Specifications | Roll and Pitch Controllers |
|---|---|
| settling time | ≤0.1 s |
| overshoot | ≤10% |
| steady-state error | zero |

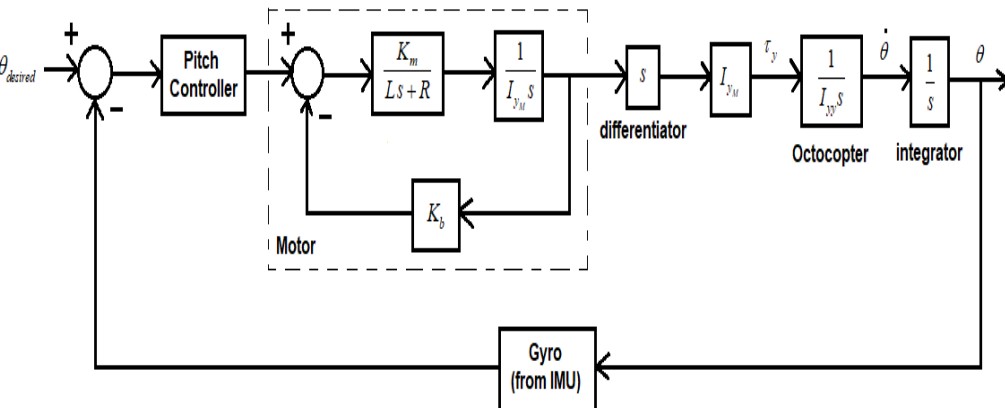

**Figure 10.** Pitch attitude control loop.

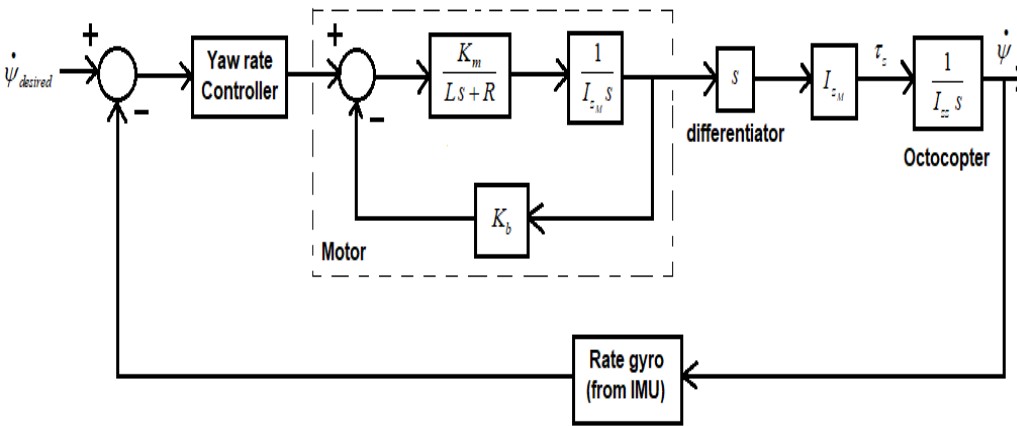

**Figure 11.** Yaw rate control loop.

### 5.4.1. Roll Attitude Controller

From Figure 9, by including the torque from the three motors, and considering the value of the gyro gain as 0.86 volts/rad and the values of the moment of inertia of the selected propeller and the octocopter in the *x*-axis, the open-loop transfer function of the roll control loop without the controller is given as follows:

$$\left( \frac{\phi}{\phi_{\text{desired}}} \right)_{\text{open\_loop}} = \frac{22.7 \times 10^{-8}}{s[(1.6 \times 10^{-11})s^2 + (1.1123 \times 10^{-7})s + 6.036 \times 10^{-6}]}. \tag{24}$$

The open-loop poles of the roll control system are $0$, $-54.7$, and $-6897.2$, and there is no zero from the transfer function. The dominant poles for the desired settling time of 0.1 s with 10% overshoot can be obtained from second-order under-damped relations and were found to be $-40 \pm 54.6$.

The open-loop transfer function obtained from the roll attitude control system is a Type-1 system. As such, the steady-state error is always zero for a step input. Therefore, integral control is not required in the roll-control loop, and the roll-angle controller is a PD controller. The location of the PD controller's zero can be obtained using the angle criterion of the root locus for the desired dominant pole. In this study, it was found to be $-177.8$.

The designed transfer function of the PD controller is therefore K(s + 177.8), where K is the gain corresponding to a 10% overshoot. From the root locus of the PD-compensated system for the roll control shown in Figure 12, the value of K in this study was 12.6. By comparing the obtained transfer function with the standard form of the PD controller, the derivative gain KD = 12.6, and the proportional gain KP = 2240.28. The octocopter response with the roll controller for the desired roll-attitude command was obtained using MATLAB,

as shown in Figure 13. From the MATLAB result, it can be seen that the setting time is less than 0.1 s, with an overshoot of approximately 10%.

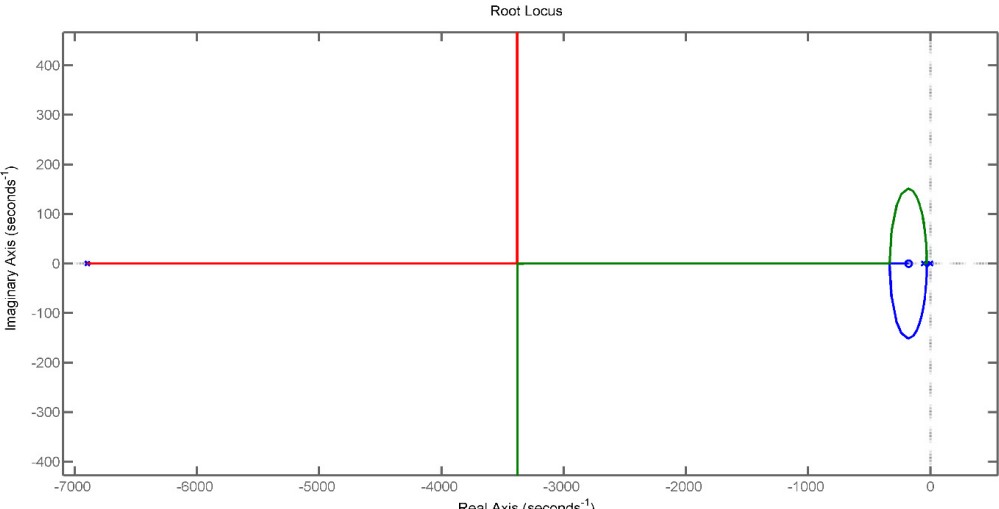

**Figure 12.** Root locus for the roll attitude-control with the designed controller.

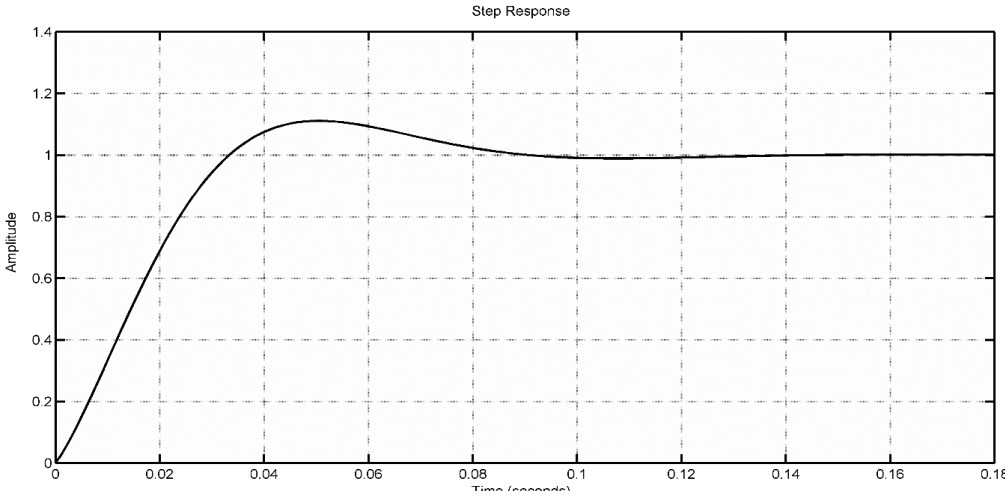

**Figure 13.** Roll response of the octocopter with the designed proportional-derivative (PD) controller.

5.4.2. Pitch Attitude Controller

From Figure 10, by including the torque from the three motors and considering the values of the moment of inertia of the selected propeller and octocopter along the *y*-axis and the value of the gyro gain, the open-loop transfer function of the pitch control loop without the controller is given as follows:

$$\left(\frac{\theta}{\theta_{\text{desired}}}\right)_{\text{open\_loop}} = \frac{16.37 \times 10^{-7}}{s\left[(1.226 \times 10^{-10})s^2 + (8.53 \times 10^{-7})s + 6.418 \times 10^{-6}\right]}. \tag{25}$$

From the transfer function, the open-loop poles of the pitch control system are 0, −7.53, and −6950, and there is no zero. The pitch-attitude system is also a Type-1 system. Therefore, a PD controller is sufficient for pitch attitude control.

Using the same procedure as in Section 5.4.1, the location of the PD controller's zero for the pitch control loop was found to be –62.7. The value of the forward loop gain corresponds to a 10% overshoot from the root found to be 37.2. Therefore, from the standard form of the transfer function of the PD controller, KP = 2332.44, and KD = 37.2. However, with the designed PD controller using the root locus approach, we observed an overshoot

of approximately 22% in the octocopter pitch response. Hence, we decided to design a PD controller by manually tuning the gains. With the manual selection of controller gains, the pitch response of the octocopter for the desired pitch-attitude command was obtained using MATLAB, as shown in Figure 14.

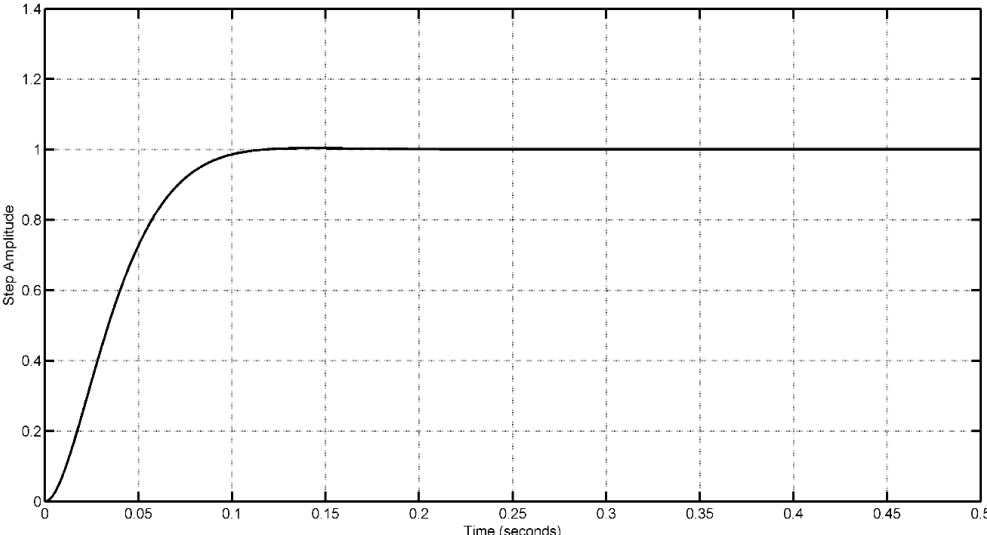

**Figure 14.** Pitch response of the octocopter with the designed PD controller.

### 5.4.3. Yaw-Rate Controller

From Figure 11, by including the torque from the four motors for creating an unbalanced torque about the *z*-axis and considering the values of the moment of inertia of the octocopter and selected propeller in the *z*-axis, the open-loop transfer function of the yaw rate-control loop without the controller by taking the rate gyro gain as 0.86 volts/(rad/s) is given as follows:

$$\left(\frac{\dot{\Psi}}{\dot{\Psi}_{desired}}\right)_{open\_loop} = \frac{20.915 \times 10^{-7}}{[(1.524 \times 10^{-10})s^2 + (1.06 \times 10^{-6})s + 8.328 \times 10^{-6}]} . \tag{26}$$

The open-loop transfer function obtained from the yaw-rate control system is a Type-0 system. Therefore, a steady-state error exists for step input. Thus, the yaw rate controller's function is to obtain a faster response with zero steady-state error. Therefore, in this study, the selected yaw-rate controller was set as a proportional-integral (PI) controller. Here, the integral action is used to obtain the zero steady-state error, and proportional action is used to obtain a faster transient response. It was desirable to obtain the response of the octocopter without any overshoot for a desired yaw rate command to maintain further stability. Based on this, the desired controller specifications for the yaw-rate command are listed in Table 3.

**Table 3.** Yaw rate controller design specifications.

| Specifications | Value |
| :---: | :---: |
| settling time | 0.005 s |
| overshoot | zero |
| steady-state error | zero |

The PI controller required a pole to be placed at the origin to increase the system's type number by one. To retain the transient response, zero was placed close to the origin. Here, the PI controller's zero was selected as 0.01, and the transfer function of the PI controller

was given by $\frac{K(s+0.01)}{s}$, where K is the forward-loop gain corresponding to zero overshoot and a settling time of 0.005 s.

The pole location corresponded to a 0.005 s settling time of $-800$. From the root locus of the yaw rate control system with the PI controller, the value of K corresponding to the pole location, 800, was found to be 355. From the standard form of the transfer function of the PI controller, the proportional gain KP = 355 and integral gain KI = 3.55. The octocopter response with the yaw-rate controller for the desired yaw-rate command was obtained using MATLAB, as shown in Figure 15.

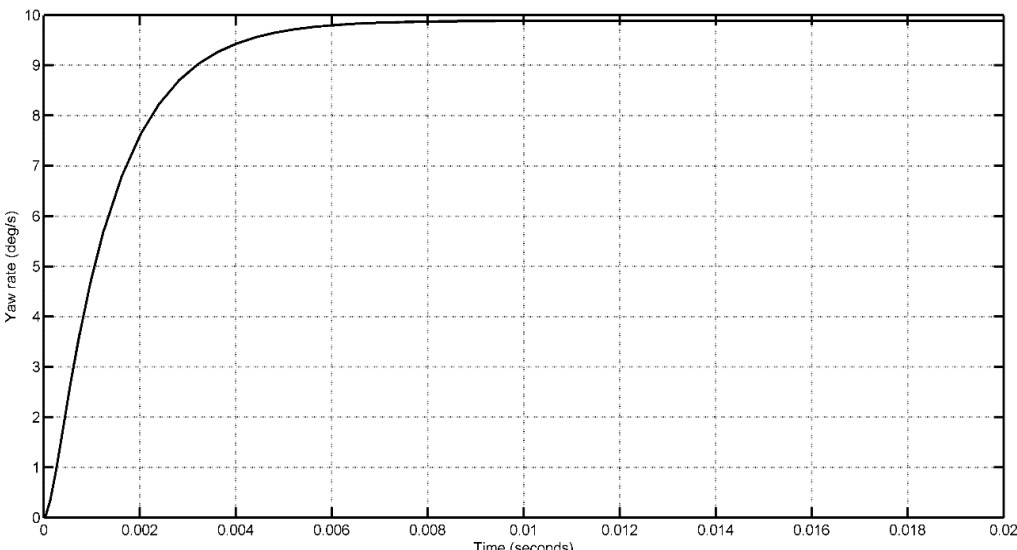

**Figure 15.** Yaw rate response of the octocopter with the designed proportional-integral (PI) controller for an input rate of 10 deg/s.

The controller gains are given in Table 4, and their responses are compared with and without the controllers in Table 5. From the comparison results, it is clear that the designed controllers satisfy our requirements.

**Table 4.** Controller gains.

| Controller | Gains | | |
|---|---|---|---|
| | Proportional Gain, KP | Integral Gain, KI | Derivative Gain, KD |
| roll controller | 2240.28 | - | 12.6 |
| pitch controller | 1105 | - | 38 |
| yaw-rate controller | 355 | 3.55 | - |

**Table 5.** Comparison of octocopter responses with and without controller.

| Performance Specifications | Roll Response | | Pitch Response | | Yaw Rate Response | |
|---|---|---|---|---|---|---|
| | Without Controller | With Controller | Without Controller | With Controller | Without Controller | With Controller |
| Settling time | 104 s | 0.0811 s | 14.9 s | 0.09 s | 0.398 s | 0.006 s |
| Rise time | 58.4 s | 0.0244 s | 8.32 s | 0.13s | 0.223 s | 0.0029 s |
| Overshoot | zero | 11% | zero | 0.4% | zero | zero |
| Steady-state error | zero | zero | zero | zero | 79.9% | zero |

## 6. Computational Fluid–Structural Results and Discussions

The next methodology involved in this fog clearance drone was CFD, including an aerostatic atmospheric analysis over the octocopter in typical and cold environments. The ultimate objective of this CFD implementation was to estimate the aerodynamic pressure and velocity variations over the octocopter and their main effects, such as the mixing behavior of the atmospheric fluid [foggy] flow and energized flow from the octocopter's storage tank, induced velocity rate below the eight propellers, and energized seawater behavior just below the nozzle section of the tank. In addition to the CFD, the MRF was additionally involved in this aerodynamic environmental analysis to represent the rotodynamic effect of the octocopter propellers. In addition to the CFD–MRF combo, an FSI-based numerical simulation was also implemented in this multidisciplinary investigation to estimate the structural analysis of the octocopter frame in cold environments. Subsequently, one-way coupling-based FSI analysis was imposed on the environmental octocopter for various working conditions, and thereafter, a suitable lightweight material was chosen based on the low reactance of structural outcomes [24–28].

### 6.1. Computational Model

The flow over the octocopter and its aerodynamic effects were significant; therefore, external CFD analyses were used in this investigation. An artificial control environmental volume was formed over an octocopter of size 6000 mm long and 3000 mm in diameter. A detailed pictorial representation is presented in Figure 16. A cylindrical shape was used for the external control volume. Generally, the main object, the octocopter in this case, plays a predominant role in the computational model description because the computational model is a fundamental platform for all kinds of simulation.

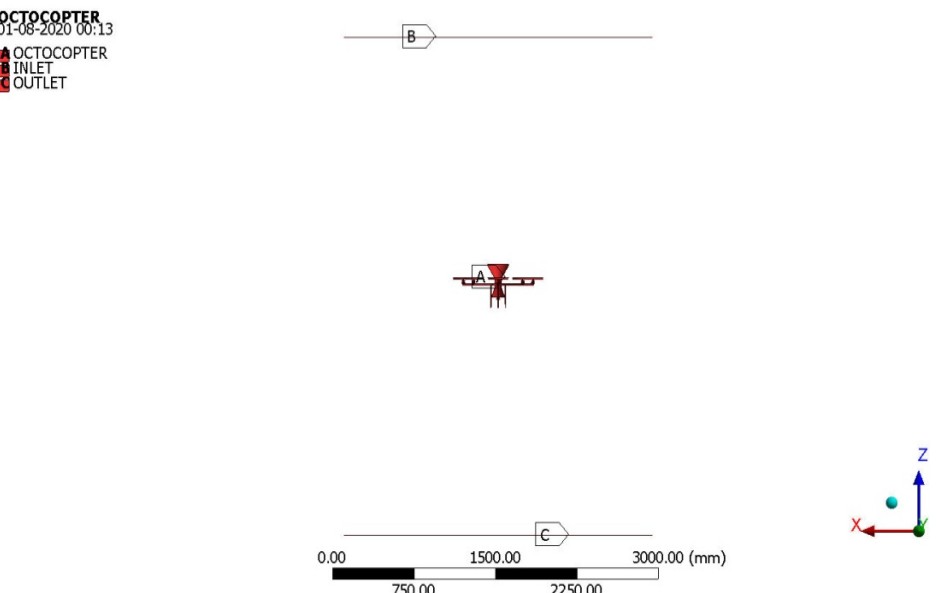

**Figure 16.** External control volume with octocopter.

### 6.2. Discretization

Unstructured, small finite volumes were constructed from the previously mentioned control environmental volume. The grid structure is illustrated in Figure 17. A refinement facility was employed over the octocopter to form high-resolution grids near the regions of the octocopter. A proximity facility was applied to represent the area variations inside the control volume, and a curvature facility was applied to represent the curved shapes inside the control volume. Owing to these good facility implementations, high-resolution grids were formed, and the value obtained was 0.98876. Additionally, grid-convergence tests were conducted to enhance the reliability of the outcomes. The first grid-convergence test

(GCT) was conducted on the fundamental factors corresponding to the induced velocity of the aerodynamic computation. In this first GCT, a total of six mesh cases were imposed: fine mesh, fine with proximity, fine with curvature, and fine with face mesh on the octocopter, fine with inflation set-up on the octocopter, and fine with inflation set-up on the entire set of components. The comprehensive result is shown in Figure 18, in which mesh Case IV was selected as the best performer. Comparatively, case IV provided reliable outcomes with lower elemental consumption.

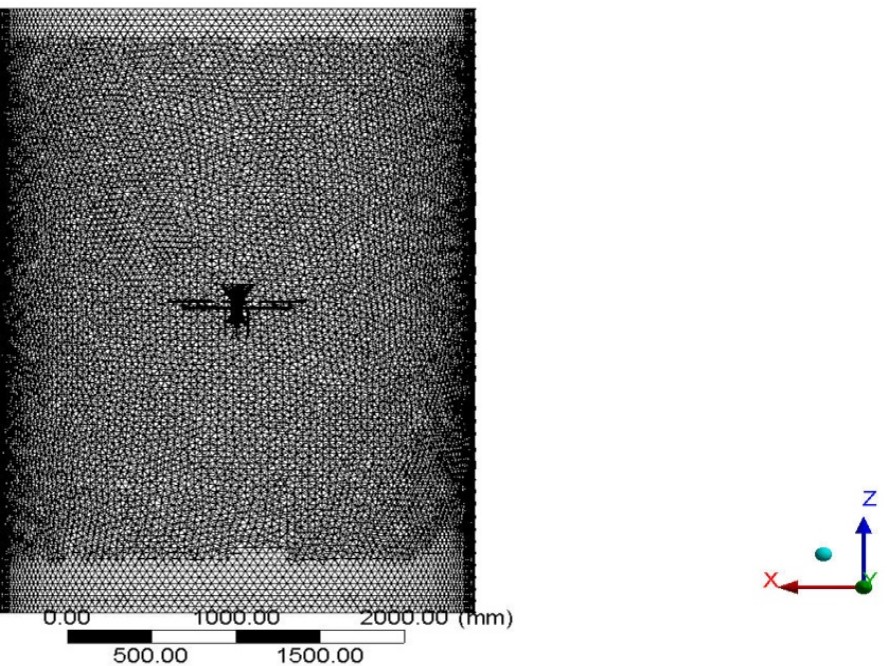

**Figure 17.** Discretized structure of the entire computational model.

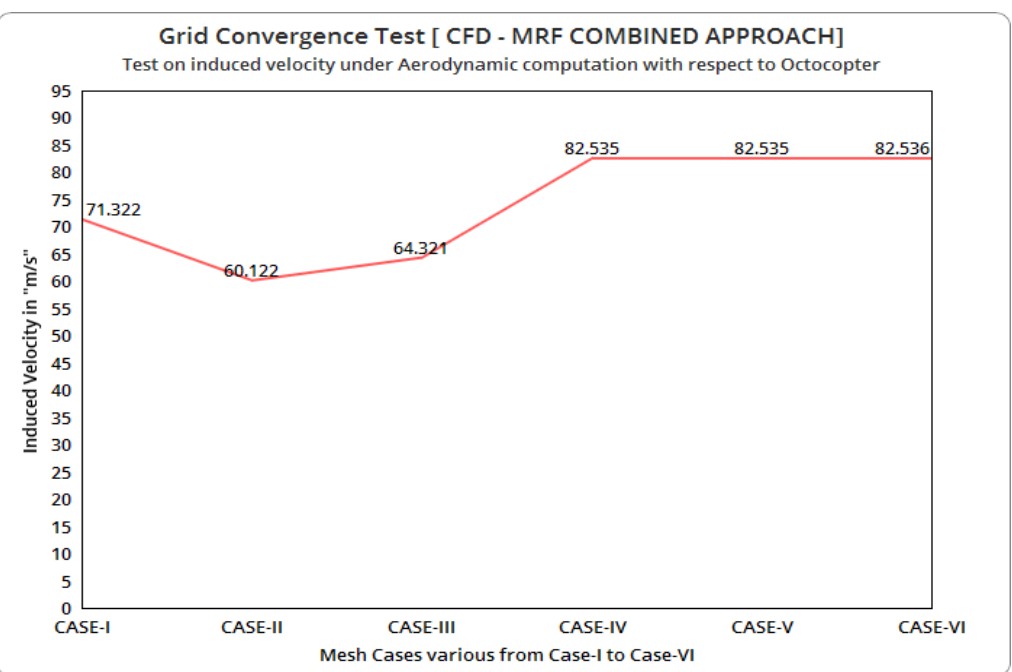

**Figure 18.** Grid-convergence results for aerodynamic analysis.

In the second GCT, six different mesh cases were imposed: coarse mesh, medium mesh, fine mesh, and fine with face on mesh on the octocopter, fine with inflation mesh

set-up on the octocopter, and fine with inflation mesh set-up on the entire control volume. The entire total deformation-based test on the GFRP is shown in Figure 19, wherein mesh Case 3 was chosen as a reliable mesh case.

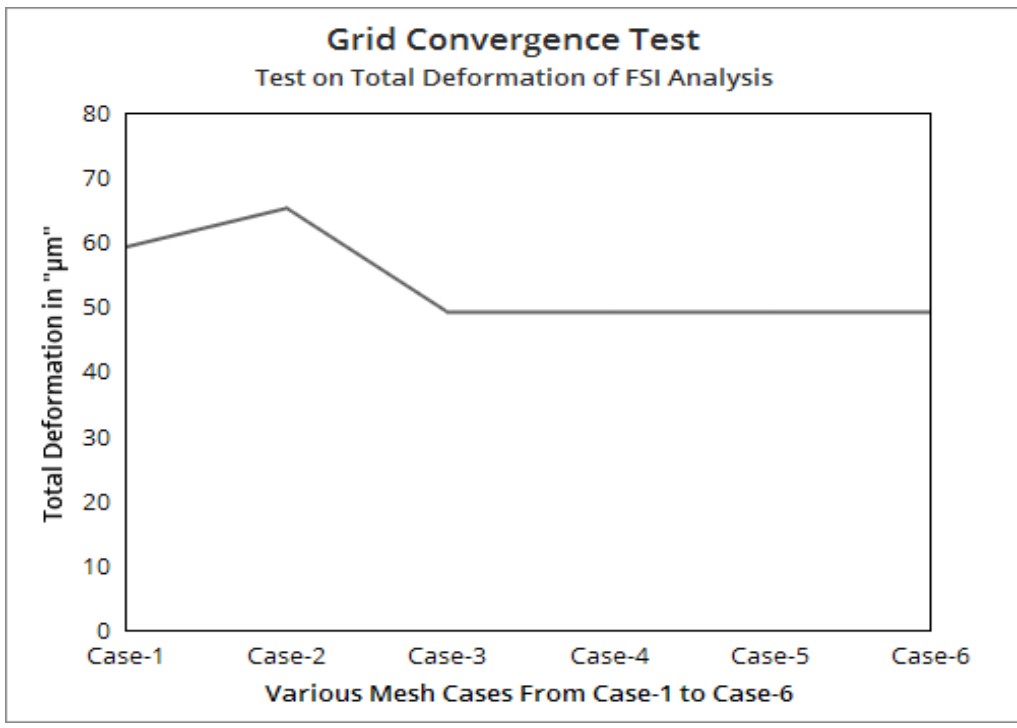

**Figure 19.** Grid-convergence results for fluid-structure-interaction (FSI) analysis.

*6.3. Boundary Conditions*

The fluid density in the environment is a predominant factor in this work, so a density-based computational solver was used as a fundamental platform. Owing to the presence of the rotors, the turbulence formation was relatively high, and thus, the k-epsilon-turbulence model was employed. Two phases were involved in these fluid analyses: a standard atmosphere and a cold (fog) atmosphere. Thus, 101,325 Pa with 1.2256 kg/m$^3$ flow properties were provided for a typical atmosphere and 105,000 Pa with 1.5 kg/m$^3$ flow properties were provided for a cold environment.

The average velocity was measured as 5.1 m/s; therefore, the same input was given as the fluid velocity for the cases. Apart from these inputs, seawater was sprayed at a velocity of 20 m/s from the octocopter to clear fog formation in locations such as airports and railway stations. The nature of this CFD analysis was under the multiflow category, and second-order derivatives were implemented in all the solution methods. The pressure and velocity couplings are very important in the execution of attainment of convergence thus a coupled scheme was used for the fluid property interactions. The entire boundary conditions are shown in Figure 20, wherein the representation of fixed and rotating control volumes provides information about the location of the walls.

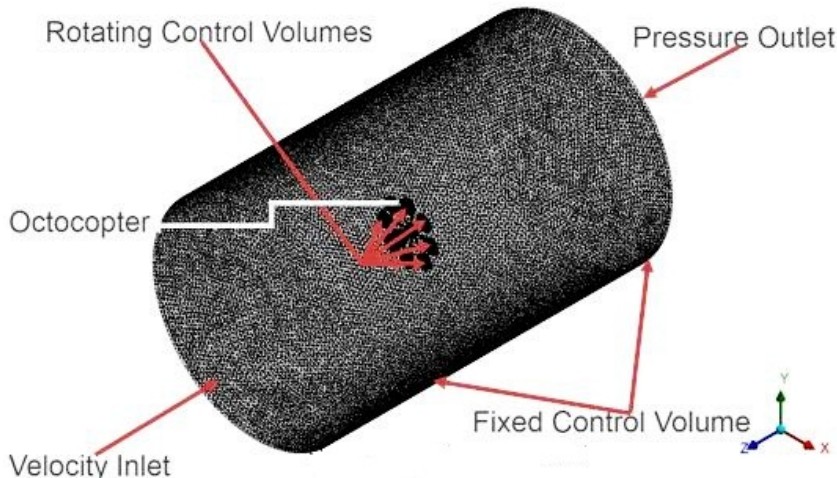

**Figure 20.** Integrated FVM modeled control volume with the component names.

### 6.4. Computational Aerostatic Results

The aerostatic analyses were computed for both cases under the abovementioned boundary conditions. The predominant outcomes of these computations are the fluid pressure on the octocopter and the fluid velocity over the octocopter. The normal atmospheric outcomes are shown in Figures 21 and 22, respectively. Figures 23 and 24 are show the aerodynamic pressure and velocity variations under the imposed foggy environmental conditions. This first-phase CFD analysis was used to predict the flow properties in environments with the presence of the octocopter and aerodynamic load-acting details on the UAV. Both these inputs are useful for attaining the objectives of this work.

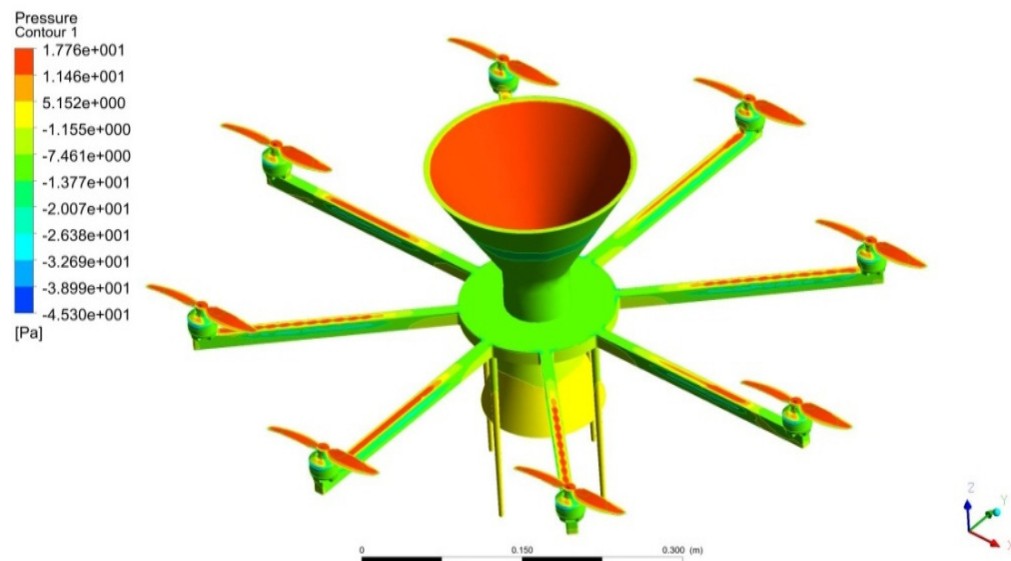

**Figure 21.** Pressure variations on the octocopter (normal atmosphere).

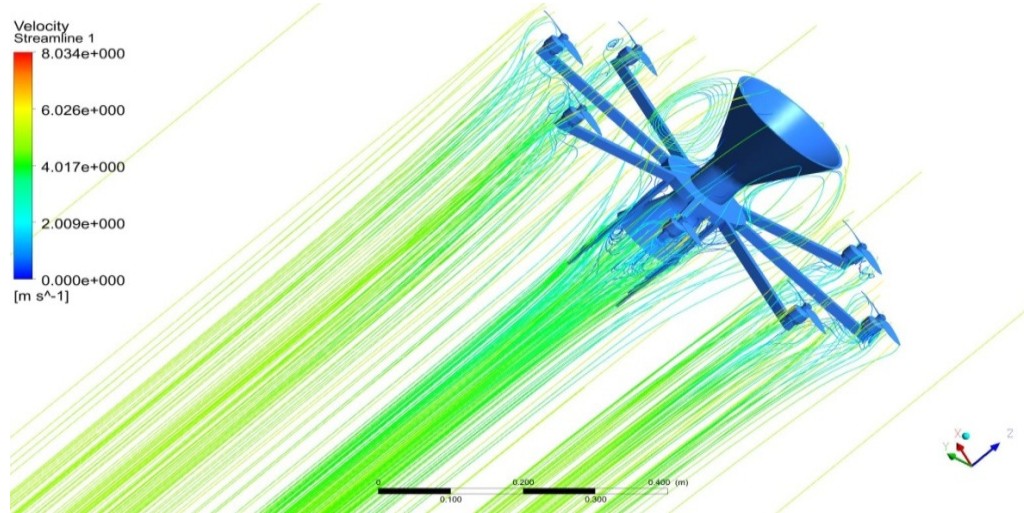

**Figure 22.** Distribution of velocity over the octocopter (normal atmosphere).

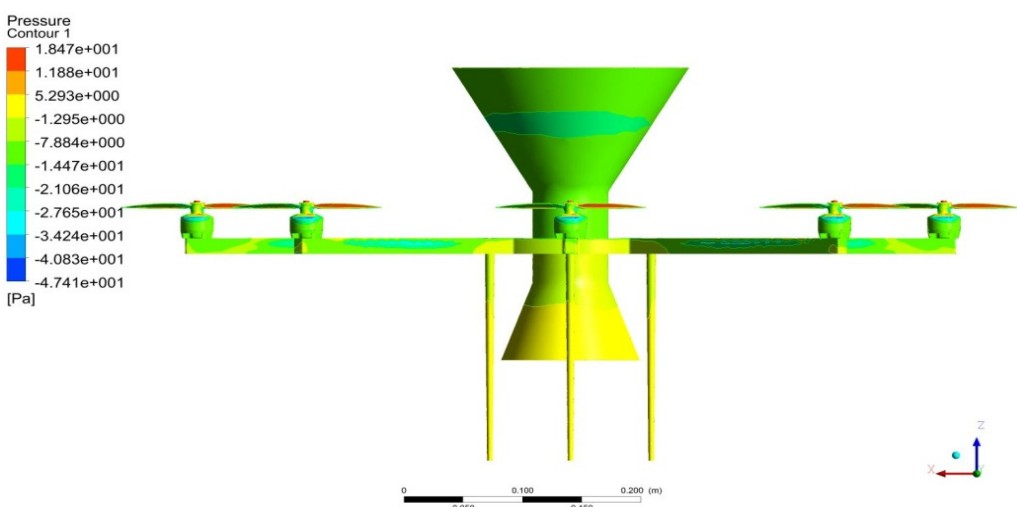

**Figure 23.** Dynamic pressure variations in foggy environment on the octocopter.

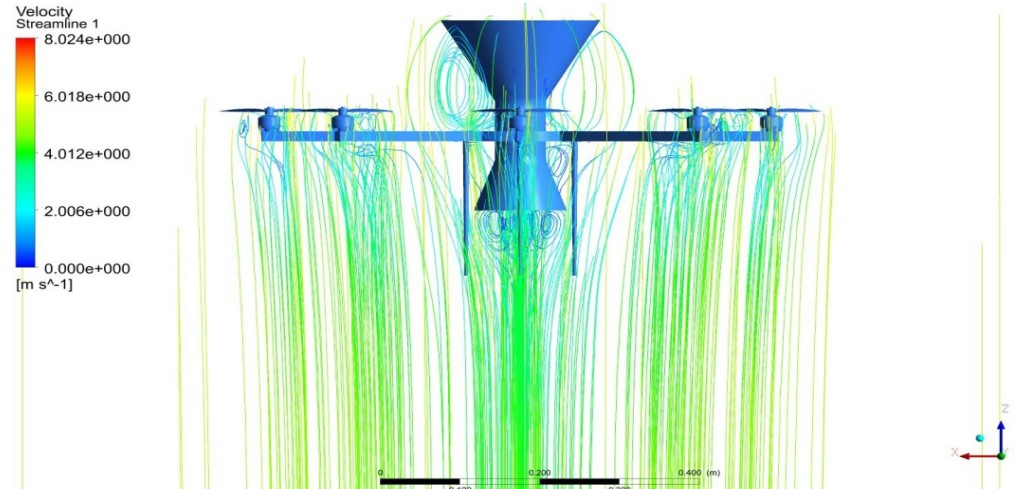

**Figure 24.** Aerostatic velocity variations over the octocopter in foggy conditions.

### 6.5. Computational Aerodynamic Results in Foggy Environments

The successful completion of the aerostatic computation increased the strength to extend the simulation for aerodynamic environmental conditions using CFD–MRF collaborative approaches. In this transient flow analysis, two different control volumes were used: a fixed control volume in a cylindrical shape with the same dimensions as mentioned above, and rotating control volumes over the eight propellers. Figures 25–28 show the pressure and velocity variations in the presence of the octocopter.

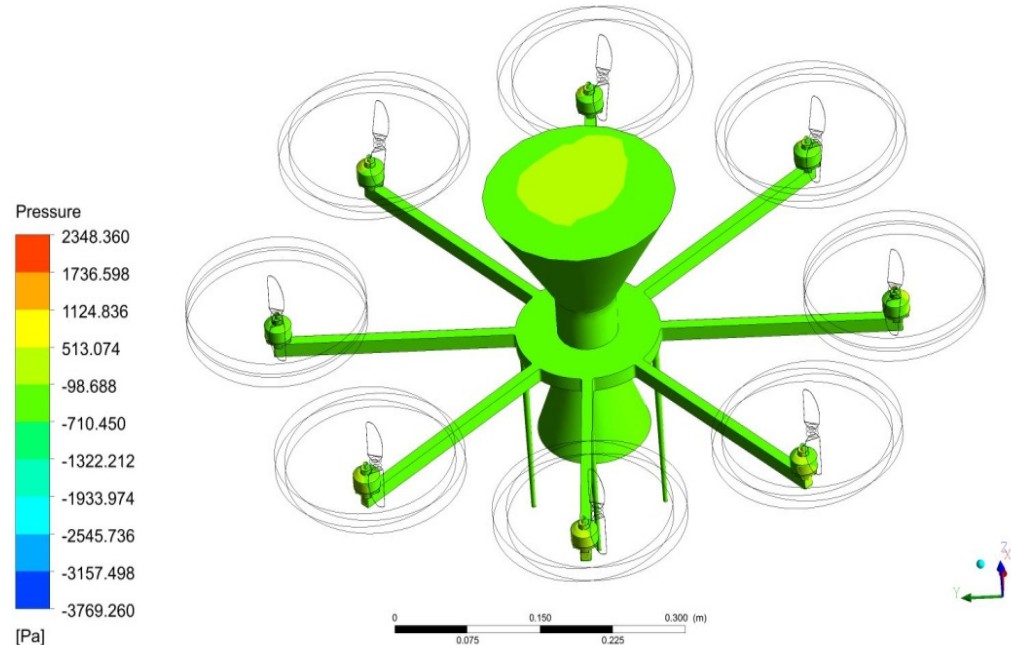

**Figure 25.** Pressure variations on the octocopter's frame.

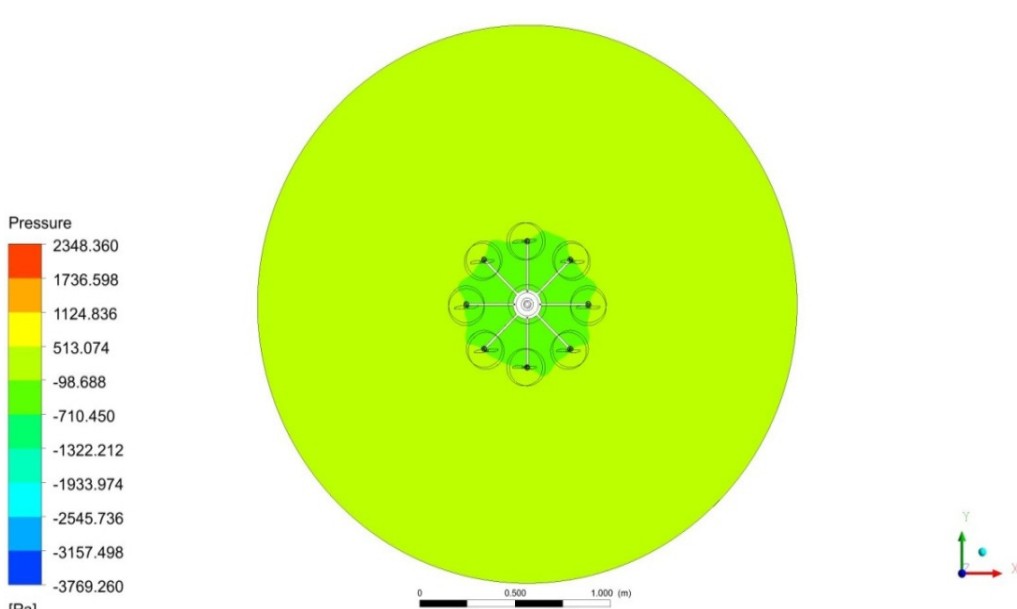

**Figure 26.** Pressure variations on the octocopter's propellers.

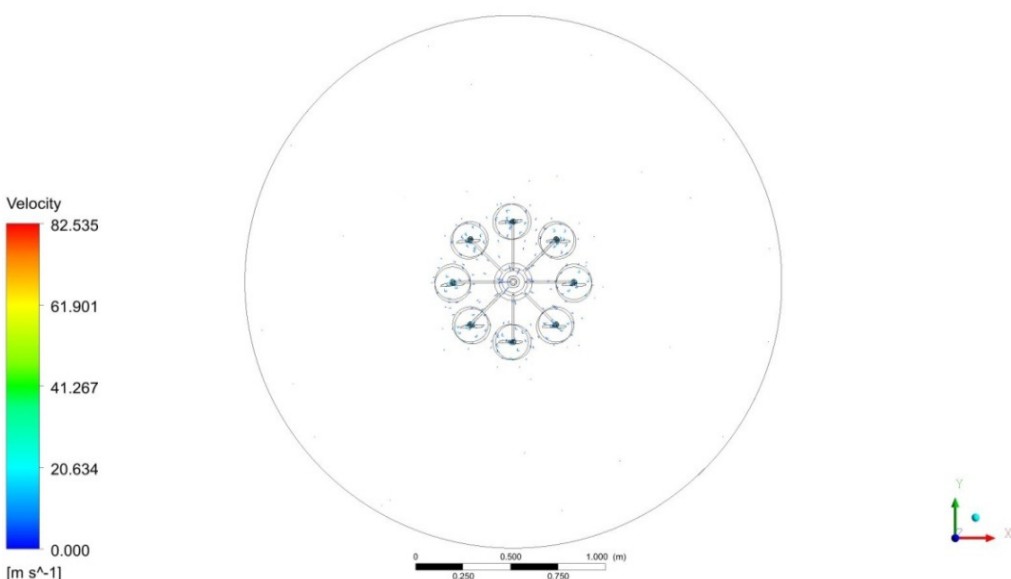

**Figure 27.** Velocity distributions over the octocopter's propellers.

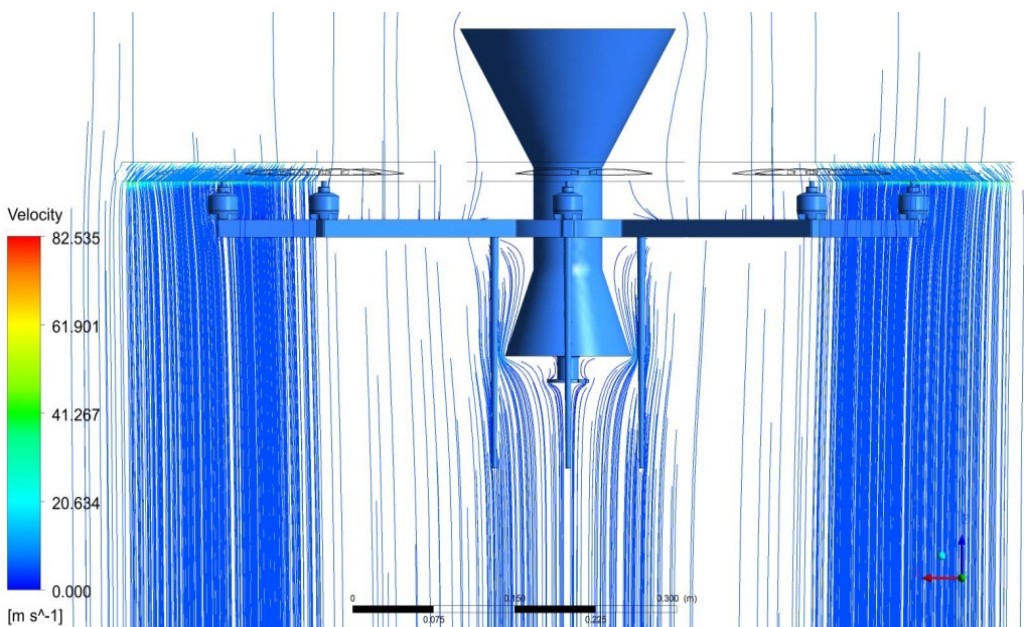

**Figure 28.** Velocity distributions over the entire fog-clearance drone.

The same boundary conditions were extended for this aerodynamic analysis, in which the RPM of the propellers and solution controls were added. The propeller's RPM was estimated to be 4412, using the standard analytical formula, and the same was applied to all rotating frames. Second-order explicit methods have been applied to this complicated aerodynamic computation.

Owing to the rotodynamic effect of seawater, the fluid velocity inside the control volume was increased by 16.2 from the normal fluid velocity; therefore, this increment can collapse the denser air particles in the fog environment. Due to these collisions, fog clearance can occur under difficult atmospheric conditions. In addition, the static pressure inside the control volume was reduced, further reducing the foggy fluid density. Therefore, the proposed multiflow analysis-based complex system is a better solution for providing a clear environment in foggy environments.

### 6.6. Fluid-Structure Interaction (FSI) Results

Owing to the high density of the fluid, a cold environment can affect the octocopter structure; therefore, to react to structural effects, FSI analyses were carried out. A one-way coupling-based FSI was implemented in this investigation, and the aerodynamic load acting on the octocopter is shown in Figure 29.

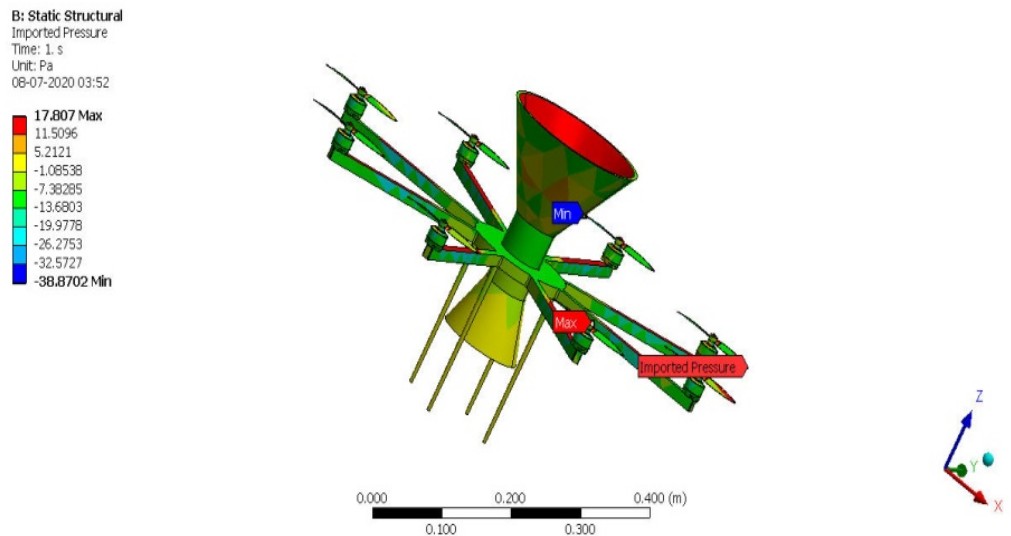

**Figure 29.** Import load from aerodynamic pressure.

The predicted aerodynamic load was primarily used as an external structural load under a uniformly distributed load category. The hinged support is provided at the bottom of all landing sticks of the octocopter. This FSI analysis was carried out for three primary materials: aluminum alloy, Epoxy-S-glass fiber-based composite material, and epoxy–carbon–woven wet-based composite material.

The structural results of the GFRP-loaded octocopter are shown in Figures 30–35. In addition to individual variations, the comparative variations in the primary structural effects are shown in Figures 36–38. Here, the following parameters are noted: aluminum alloy is better than the other materials, from a structural perspective. The GFRP composite is the next-best performer after aluminum alloy, but its weight is double that of the alloy. Therefore, GFRP is the best material for use in environmental applications.

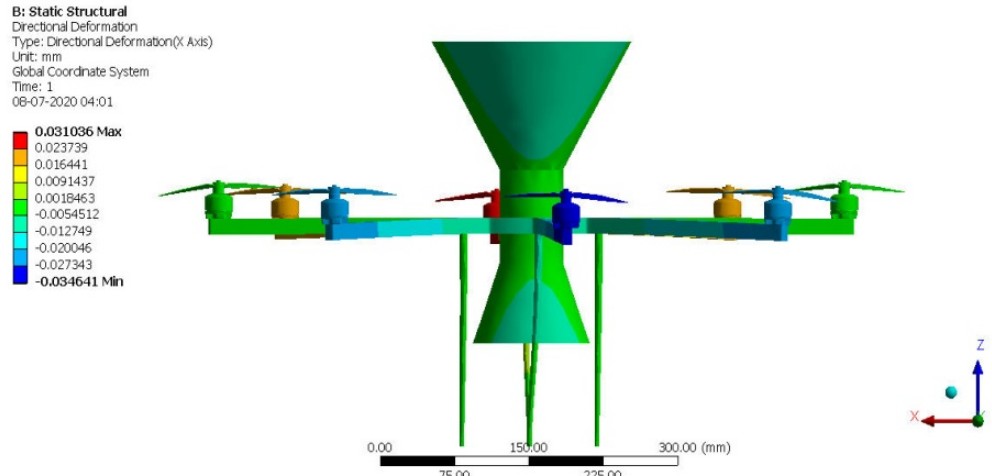

**Figure 30.** Variations of displacement in the X direction.

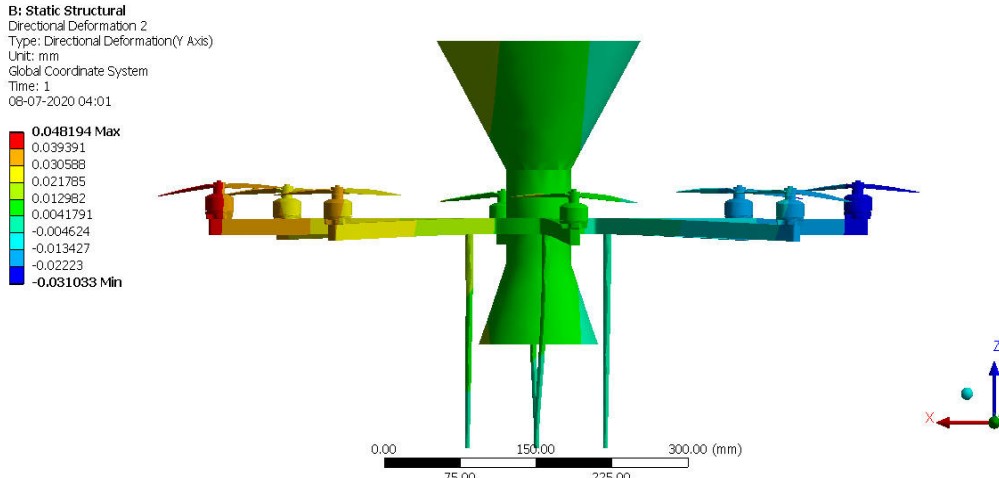

**Figure 31.** Variations of displacement in the Y direction.

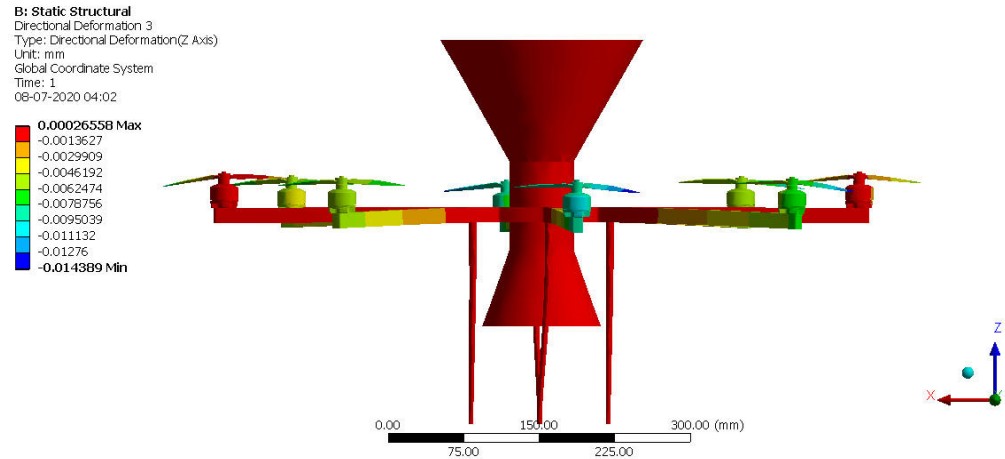

**Figure 32.** Variations of displacement in the Z direction.

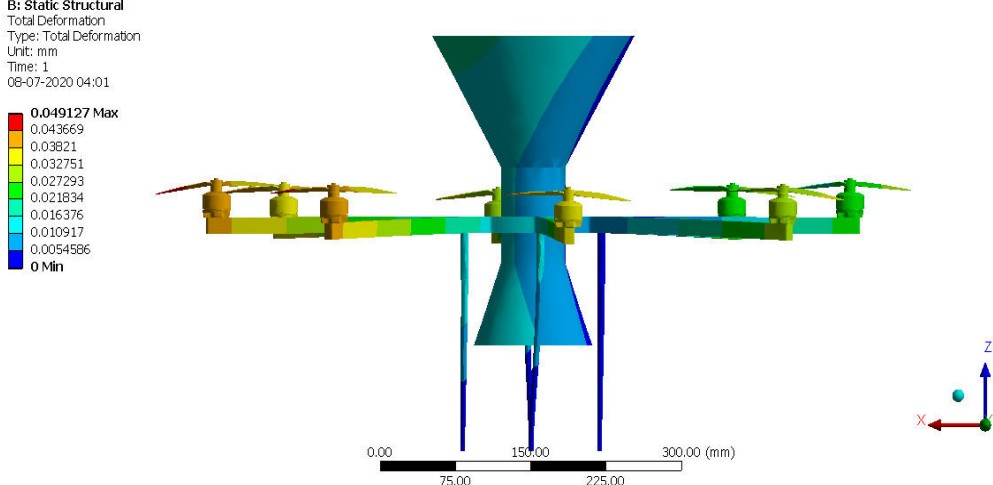

**Figure 33.** Total deformed structure of the octocopter.

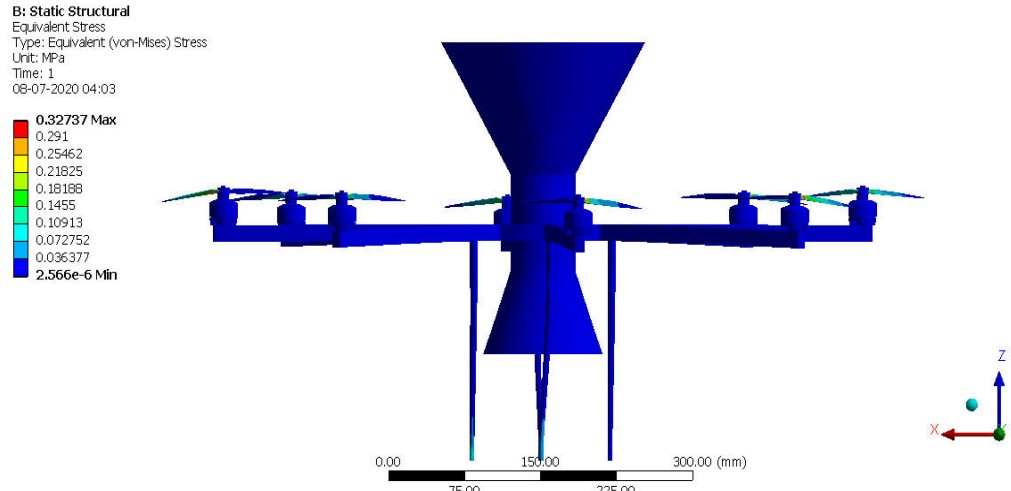

**Figure 34.** Variations of equivalent stress on the octocopter.

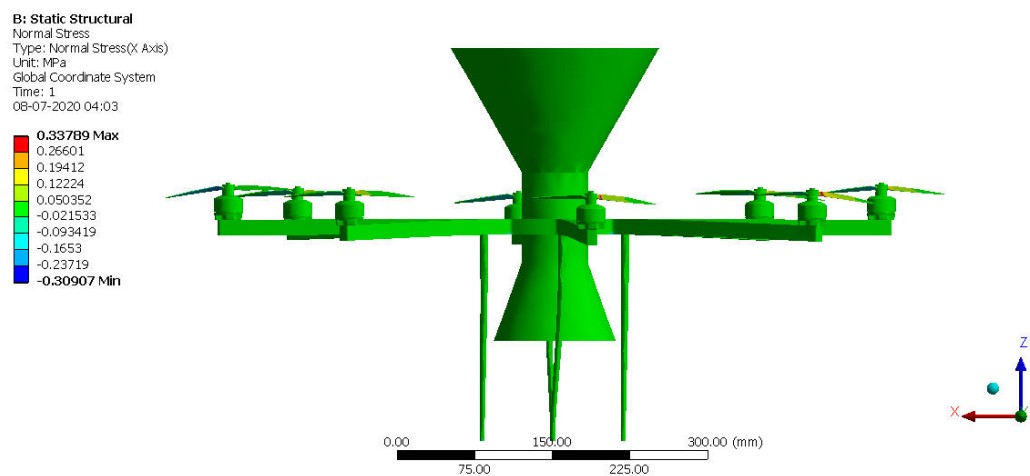

**Figure 35.** Variations of normal stress on the octocopter.

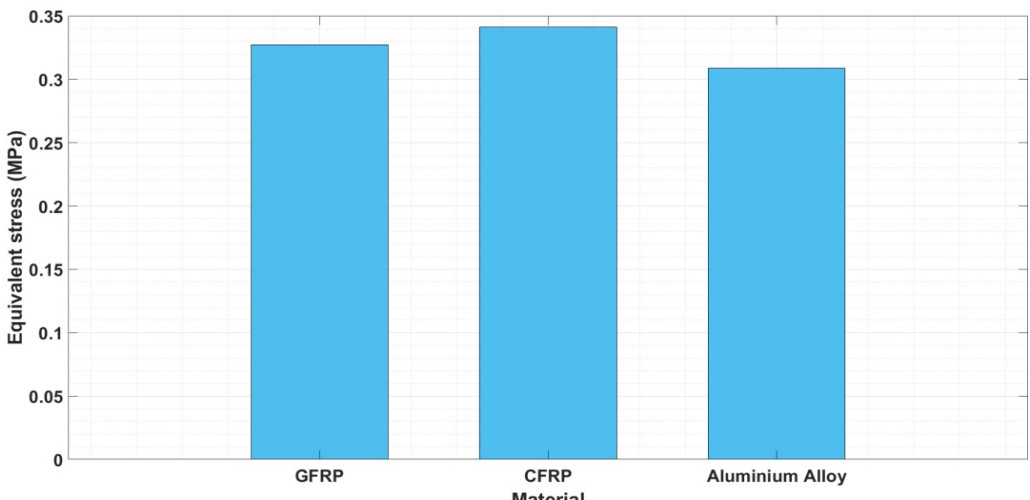

**Figure 36.** Comparative equivalent stress variations on the primary aerospace materials (including glass-fiber-reinforced polymer (GFRP) and carbon-fiber-reinforced polymer (CFRP).

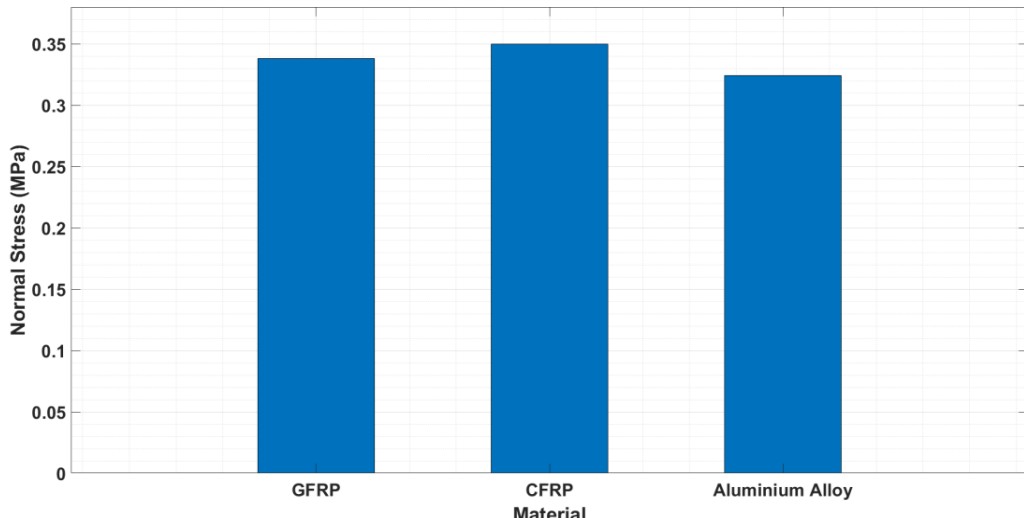

**Figure 37.** Comparative normal stress variations on the primary aerospace materials.

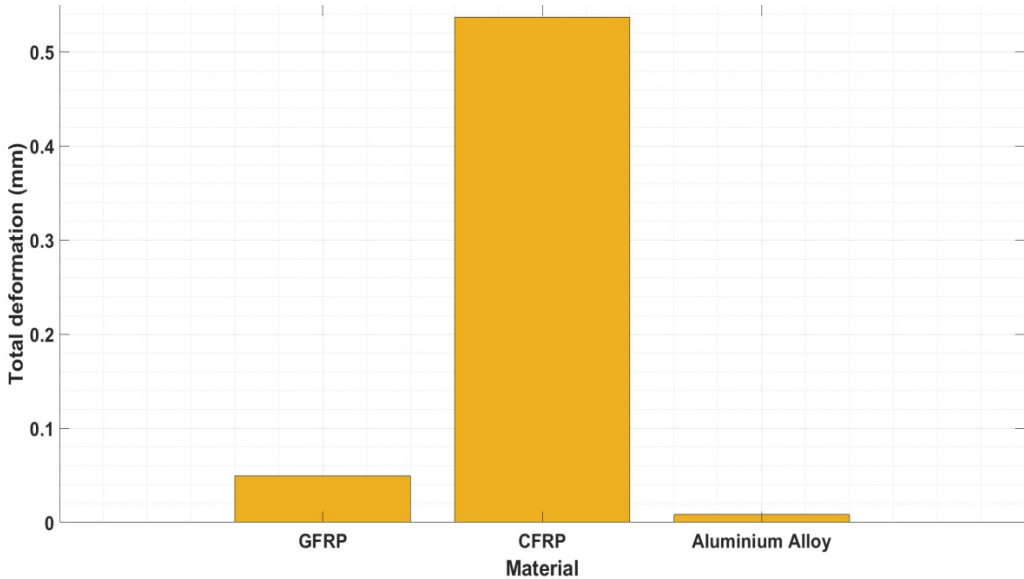

**Figure 38.** Comparative deformation variations on the primary aerospace materials.

*6.7. Discussions*

The predominant observations in both steady- and transient-flow analyses of the CFD simulations are turbulence formations after the octocopter and its rotors. The presence of a high amount of rotodynamic nature-based components and, according to bluff-body aerodynamics [top position of storage tank], the wake creations are much higher than other conventional multirotor UAVs, which are also verified through CFD outcomes [refer Figures 22, 24 and 28]. The high wakes provide a platform for complete flow mixing between the energized fluid and foggy/smokey fluid. Because of this turbulence, the kinetic energy of the fluid has been increased compared to the previous higher level, which completely pushes the foggy fluid toward groundward. Therefore, three effects (C–D duct effect, rotodynamic effect of eight rotors, and high turbulence) provide a high amount of kinetic energy to the ejector from the tank. Owing to this high kinetic energy, the highly dense seawater dominates the foggy fluid movement and forces the foggy fluid towards the ground. The pressure results [Figures 23, 25 and 26] are only included in the dynamic pressure to estimate the exact pressure on the octocopter, and the operating pressures need to be added. The major factors involved in the selection of a suitable material to resist under foggy aerodynamic loading conditions are low deformation and low induction of

stresses. The maximum stresses are induced at the level of 0.35 MPa and the maximum deformation generated by the octocopter's structure is 0.40 mm. Both of these values are significantly lower than the ultimate load of the selected lightweight materials; therefore, the imposed octocopter is structurally stable without any conditions and has a safety factor of more than 100 under these critical environmental conditions.

## 7. Conclusions

Environmental applications such as fog clearance, air-pollution control, and smoke clearance need to be solved using an advanced system; accordingly, in this study, the use of an octocopter is proposed. This work's predominant contribution is a unique octocopter design. Standard analytical formulae are used to estimate the conceptual design parameters. The design of the arm, design of the landing stick, and payload design play an essential role in the analytical estimation. The aerodynamic performance parameters, such as the lift and drag coefficients, were estimated for both vertical-climb and forward-speed operations using standard analytical methods. The attitude dynamics of the octocopter were introduced with a linearized model along with the motor dynamics, and the controller-design and -stability analyses were performed for attitude control. The simulation results gave a faster response from the octocopter with acceptable overshoot and zero steady-state error. In addition, we suggest the implementation of sensor-fusion algorithms to improve the performance of the control system, owing to the presence of gyro errors. An advanced CFD tool-based methodology was implemented for the estimation of the aerostatic fluid property variations over the octocopter. The coupled computation [CFD–MRF] is involved in estimating the aerodynamic properties in intended foggy environments, in which the fluid properties are monitored. The fluid velocities are increased by multiples of 16.2, and the static pressures are decreased, which can help manage foggy environments by reducing the foggy-fluid density. Three effects (C-D duct effect, rotodynamic effect of eight rotors, and high turbulence) imparted a large amount of kinetic energy to the fluid from the exit of the tank. Owing to this large kinetic energy, the highly dense seawater dominates the foggy fluid movement and forces the foggy fluid towards the ground. Finally, FSI analyses were performed for the three necessary composite materials, and the GFRP-based composite was determined to be the most suitable. Through FSI, it was strongly observed that the environment does not affect the structural parts of the octocopter because the induced stress values (0.40 MPa) are 250 times lesser than the ultimate stresses of the implemented lightweight materials. Thus, the proposed octocopter is structurally stable and can be efficiently used in many types of environmental applications.

**Author Contributions:** Conceptual Design, V.R. and P.R.; Modeling and Control, S.K.S. and S.J.; CFD and FSI, V.R. and S.K.M.; writing—original draft preparation, S.J. and P.R. All authors have read and agreed to the published version of the manuscript.

**Funding:** Following are results of a study on the "Leaders in INdustry-university Cooperation +" Project, supported by the Ministry of Education and National Research Foundation of Korea (No. 2021-C-G031-010109).

**Institutional Review Board Statement:** Not applicable.

**Informed Consent Statement:** Not applicable.

**Data Availability Statement:** Not applicable.

**Conflicts of Interest:** The authors declare no conflict of interest.

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
