# Peer review of "Conceptual Design and Multi-Disciplinary Computational Investigations of Multirotor Unmanned Aerial Vehicle for Environmental Applications"

_applsci, doi:10.3390/app11188364_

Round 1

Reviewer 1 Report

This study provides a conceptual design for a multi-rotor unmanned aerial vehicle (UAV); standard analytical methods are applied for parameter estimates. This UAV potentially could be used for environmental applications such as fog removal and fire suppression. The manuscript reads well in general, but the study motivation and novelty are not highlighted. Here are several major issues:

  1. Abstract section does not precisely summarize the essence of the study. For example, This sentence makes no sense logically " The Octocopter includes a storage tank that sucks the seawater and stores it, allowing it to fly in a foggy zone.". And the following sentences " The fine saltwater is sprayed from the flying vehicle and dispersed into the low clouds, thereby altering the vapor’s microphysical processes to break it up and improve visibility. The salt particles absorb the fog particles, creating an imbalance that allows for the dissipation of fog." has nothing to do with the main content of the study. And there are no results to directly support this conclusion " The results show that the probability of an accident at an airport can be decreased." .
  2. Introduction section does not have a single reference and is not informative.  Readers can not tell any pros/cons of UAV techs and applications to environmental engineering, nor the motivation of this study.
  3. Literature survey section is not sufficient and fails to tightly connected to the study's objectives, as claimed in the abstract. Lines 78-84 and 108-114 are not literature survey at all. I would suggest a fusion of introduction and literature survey, and make it very clear about the current research/application status, and then highlight the motivation, objective and novelty of this study.
  4. The rest manuscript is separated into too many sections, making it too fragmented.  Please check the Journal template (https://www.mdpi.com/journal/applsci/instructions) for a smooth logic flow of the manuscript.
  5. The meaning of each symbol in the equation should be clearly denoted right after the equation. 
  6. Some calculation processes (e.g., Lines 145, 168, 193 and many others) can be simplified or removed.
  7. Figures: please increase the dpi of some figures; be consistent with the use of 'Fig' or 'Figure' throughout the entire manuscript; please make the legends more readable by setting a constant and small decimal number.
  8. I find it very difficult to catch the discussion of the study. It is better to highlight the discussions in a separate section.
  9. Conclusions: In the end, the authors claimed that "Thus, the proposed Octocopter can efficiently solve all types of environmental applications". I would accept it if the authors can provide sufficient evidence to prove it.
  10. 16 references seem not enough to support this study.

Author Response

REVIEWER – 1

//1. Abstract section does not precisely summarize the essence of the study. For example, This sentence makes no sense logically “The Octocopter includes a storage tank that sucks the seawater and stores it, allowing it to fly in a foggy zone.". And the following sentences " The fine saltwater is sprayed from the flying vehicle and dispersed into the low clouds, thereby altering the vapor’s microphysical processes to break it up and improve visibility. The salt particles absorb the fog particles, creating an imbalance that allows for the dissipation of fog." has nothing to do with the main content of the study. And there are no results to directly support this conclusion “The results show that the probability of an accident at an airport can be decreased.”. //

Response to the comment:

The abstract has been updated with inclusion of all of the abovementioned comments.

//2. Introduction section does not have a single reference and is not informative.  Readers can not tell any pros/cons of UAV techs and applications to environmental engineering, nor the motivation of this study. //

Response to the comment:

The introduction section has been updated completely as per the above suggestions.

//3. Literature survey section is not sufficient and fails to tightly connected to the study's objectives, as claimed in the abstract. Lines 78-84 and 108-114 are not literature survey at all. I would suggest a fusion of introduction and literature survey, and make it very clear about the current research/application status, and then highlight the motivation, objective and novelty of this study. //

Response to the comment:

As per the suggestions, a literature survey has been added, and thereafter the motivation, objective, and scope of this work are mentioned.

//4. The rest manuscript is separated into too many sections, making it too fragmented.  Please check the Journal template (https://www.mdpi.com/journal/applsci/instructions) for a smooth logic flow of the manuscript. //

Response to the comment:

This paper deals with multi-disciplinary investigations (CFD, FSI, and control dynamics) of the Octocopter, so instead of three main headings, only two main headings were included in the draft. In particular, two major approaches, CFD and FSI, comprise a single section [section 6], in which different environments are investigated to clarify the fluid behaviors under different fluid conditions. This is the reason why a large number of sub-headings were used.

//5. The meaning of each symbol in the equation should be clearly denoted right after the equation. //

Response to the comment:

The meanings of all the symbols have been explained.

//6. Some calculation processes (e.g., Lines 145, 168, 193 and many others) can be simplified or removed. //

Response to the comment:

Additional calculations were performed and the results were removed. All the included calculations are very important for the design process of the Octocopter.

//7. Figures: please increase the dpi of some figures; be consistent with the use of 'Fig' or 'Figure' throughout the entire manuscript; please make the legends more readable by setting a constant and small decimal number. //

Response to the comment:

The quality of the figures has been enhanced.

//8. I find it very difficult to catch the discussion of the study. It is better to highlight the discussions in a separate section. //

Response to the comment:

A separate discussion section has been included following the CFD and FSI results.

//9. Conclusions: In the end, the authors claimed that "Thus, the proposed Octocopter can efficiently solve all types of environmental applications". I would accept it if the authors can provide sufficient evidence to prove it. //

Response to the comment:

Clear explanations and computations were carried out to investigate various environmental conditions.

//10. 16 references seem not enough to support this study. //

Response to the comment:

12 more relevant references are added.

Reviewer 2 Report

The work presented in this manuscript is related to multi-rotor unmanned aerial vehicle for environmental applications using as a research strategy a multidisciplinary approach. This work reveals a very interesting topic and approach used to investigate the main purpose of this manuscript. Please find bellow my comments about this manuscript:

  1. Abstract section, lines 24 to 30, maybe the authors may which to change the sentences sequence, i.e. first introduce how they investigate and software, and then conclude with the main observations
  2. I was expecting in the introduction section some introduction to the methods from the state of the art, which are then more described in section 2. Literature survey
  3. Since the critical factor is the weight, it is possible to use a CFD software that already has an optimization module? It was what the authors did in this study? If yes, please can you add information on Section 3
  4. Why the reason for the weight fixed at 1.25 kg, some reference from literature, some limitation from experimental devices
  5. Please, there are no need to include all the math steps, line 145 can be void if the information is more complete in line 148, for example "the overall weight was estimated as 4166.67 g considering Wpl = 1250 g." (Please check further in the manuscript for the similar calculations that can be avoid, such as line 168, 170, ...)
  6. Maybe Figure 2 can be moved to the introduction section, thus the reader will have in mind the geometry and can connect better from the very beginning with the problem under analysis
  7. I was expecting a more detailed conclusion where the results from CFD and control can be more analysed as conclusions from this work, like how they contribute to the state of the art, what can be done next
  8. I think that the CFD and the control results are presented such as separated sections, but both results can be linked, i.e. what control results are showing that agree with the CFD results and how CFD results can help on the dynamic control, CFD results have countor and streamlines in this way it is possible to infer diferent measuring points and maybe variables in order to properly define the control strategy or even help when choosing between a simple or more advanced control strategy.

Author Response

REVIEWER – 2

//1. Abstract section, lines 24 to 30, maybe the authors may which to change the sentences sequence, i.e. first introduce how they investigate and software, and then conclude with the main observations//

Response to the comment:

The abstract section has been modified and updated as per the suggestions given.

//2 & 3. I was expecting in the introduction section some introduction to the methods from the state of the art, which are then more described in section 2. Literature survey

Since the critical factor is the weight, it is possible to use a CFD software that already has an optimization module? It was what the authors did in this study? If yes, please can you add information on Section 3. //

Response to the comment:

Detailed explanations and figures have been incorporated in the sections mentioned. 

//4. Why the reason for the weight fixed at 1.25 kg, some reference from literature, some limitation from experimental devices//

Response to the comment:

The compositional elements involved in the payload weight are included, and explanations of the primary and secondary weight contributions are included.

//5. Please, there are no need to include all the math steps, line 145 can be void if the information is more complete in line 148, for example "the overall weight was estimated as 4166.67 g considering Wpl = 1250 g." (Please check further in the manuscript for the similar calculations that can be avoid, such as line 168, 170, ...) //

Response to the comment:

The calculation steps have been checked, and the additional calculations removed.

//6. Maybe Figure 2 can be moved to the introduction section, thus the reader will have in mind the geometry and can connect better from the very beginning with the problem under analysis//

Response to the comment:

Figure 2 has been moved to the Introduction section

//7. I was expecting a more detailed conclusion where the results from CFD and control can be more analyzed as conclusions from this work, like how they contribute to the state of the art, what can be done next? //

Response to the comment:

The conclusions have been updated as per the suggestions given.

//8. I think that the CFD and the control results are presented such as separated sections, but both results can be linked, i.e. what control results are showing that agree with the CFD results and how CFD results can help on the dynamic control, CFD results have counter and streamlines in this way it is possible to infer different measuring points and maybe variables in order to properly define the control strategy or even help when choosing between a simple or more advanced control strategy. //

Response to the comment:

The suggestions proposed by the reviewer are valid. However, from the authors’ perspective, CFD is focused on investigating the blow behavior, turbulent formations, and induced velocity by the presence of an Octocopter and its rotor. This is the reason why the CFD and control dynamics are separated in the final draft. 

Reviewer 3 Report

The paper fits the journal scope and is written in good English. The TurnItIn antiplagiarism check showed only 14% of text similarity, which is a very good value. I enjoyed reviewing this paper, but it sounds more like an extended technical note than a scientific paper. Thus, I doubted between rejecting the article (with resubmission recommendation) or a major revision. In the end, I thought that the article can be interesting for the readers as it presents multiple practical aspects. Thus, I recommended a major revision with recommendations that are to make it a scientific paper.

1. Present the novelty (contribution of this paper to the current state of the art) at the end of section 2.

2. The literature review is referring only to six references, and this number must be greater. I would advise the authors to refer e.g. to the following:

[R1] Lei Y. , Huang Y, Wang H.“, Aerodynamic Performance of Octo-Rotor SUAV with Different Rotor Spacing in Hover”, Processes, Vol. 8(11), Paper 1364, 2020 – investigating aerodynamic performance for an octocopter

[R2] Lichota P.: "Multi-Axis Inputs for Identification of a Reconfigurable Fixed-Wing UAV", Aerospace, Vol. 7(8), Paper 113, 2020 – presenting a novel approach for UAV system identification

[R3] Chu T., Starek M.J., Berryhill J., Quiroga C.; Pashaei M.: “Simulation and Characterization of Wind Impacts on sUAS Flight Performance for Crash Scene Reconstruction”, Drones, Vol. 5(3), Paper 67, 2021. – investigating atmospheric parameters influence on UAV motion

and other MDPI papers to improve this aspect

3. Presenting calculations explicitly is not recommended and should not be done (However, the formulas used and the input values should be given explicitly)

4. Fig 2 and Fig 3 should be eliminated if Fig 1 is shown. The text should be modified accordingly.

5. In the formula for the velocity on page 10 there is a typo (dot between 1.518 and dt). However, due to comment no three, the evaluation should be eliminated.

6. Describe the coordinate system in which the inertia components are given.

7. Large overshoot for pitch response can be observed (22.3%). Would you please comment on that?

8. For the UAV and control systems, the IMU can strongly affect the performance. Provide a quantitative discussion of this aspect at least.

9. Please provide more details about the mesh e.g. element type, domain size and CFD evaluations (computational time, hardware).

10. Please add either stability and control derivatives or present all aerodynamic coefficients (i.e. Cx0, Cxu, Cxw, …) for a set of selected trim points (in a table). Was model stitching used?

11. The paper focuses on conceptual design and thus does not present results from real life application. This is fine, but it should be justified by presenting that this approach is used in other studies, e.g.: This approach is often used in flight mechanics as it reduces costs when new designs and system modifications are tested [R4 R5].

[R4] Topczewski S., Ĺ»ugaj M., Bibik P., „Impact of actuators backlash on the helicopter control during landing on the moving vessel deck”, Aircraft Engineering and Aerospace Technology, 2021

[R5] Lichota P., Noreña D. A.: "A Priori Model Inclusion in the Multisine Maneuver Design", 17th International Carpathian Control Conference, IEEE, Tatranska Lomnica, Slovakia, may 2016, pp. 440-445.

For now, this is all for me. I hope that answering those comments will help the authors to raise their paper quality.

Author Response

REVIEWER – 3

// 1. Present the novelty (contribution of this paper to the current state of the art) at the end of section 2. //

Response to the comment:

The novelty has been included at the advised position, which is after the literature section.

// 2. The literature review is referring only to six references, and this number must be greater. I would advise the authors to refer e.g. to the following:

[R1] Lei Y. , Huang Y, Wang H.“, Aerodynamic Performance of Octo-Rotor SUAV with Different Rotor Spacing in Hover”, Processes, Vol. 8(11), Paper 1364, 2020 – investigating aerodynamic performance for an octocopter

[R2] Lichota P.: "Multi-Axis Inputs for Identification of a Reconfigurable Fixed-Wing UAV", Aerospace, Vol. 7(8), Paper 113, 2020 – presenting a novel approach for UAV system identification

[R3] Chu T., Starek M.J., Berryhill J., Quiroga C.; Pashaei M.: “Simulation and Characterization of Wind Impacts on sUAS Flight Performance for Crash Scene Reconstruction”, Drones, Vol. 5(3), Paper 67, 2021. – investigating atmospheric parameters influence on UAV motion and other MDPI papers to improve this aspect //

Response to the comment:

All the suggested literature is relevant, so it has been considered and included in the references.

//3. Presenting calculations explicitly is not recommended and should not be done (However, the formulas used and the input values should be given explicitly) //

Response to the comment:

The calculation steps have been revised, and the additional calculations removed.

// 4. Fig 2 and Fig 3 should be eliminated if Fig 1 is shown. The text should be modified accordingly. //

Response to the comment:

Figure 2 has been upgraded and moved to the introduction section to enhance the clarity of the work. Figure 3 clearly shows the presence of the nozzle sprayer, which is very important; therefore, Figure 3 has been retained. The text is modified accordingly.

// 5. In the formula for the velocity on page 10 there is a typo (dot between 1.518 and dt). However, due to comment no three, the evaluation should be eliminated. //

Response to the comment:

The page has been checked, and the formula corrected. According to the authors, four different velocities have been used: velocity of foggy fluid, velocity of fluid ejected from tank, velocity of UAV, and velocity induced by the propeller.  Thus, the velocity calculation of the UAV is very important to obtain clarity in this investigation.

// 6. Describe the coordinate system in which the inertia components are given. //

Response to the comment:

The body coordinate system used to represent the forces and torques has been explained in the revised manuscript.

// 7. Large overshoot for pitch response can be observed (22.3%). Would you please comment on that? //

Response to the comment:

We observed an overshoot of approximately 22% in the pitch response for the designed PD pitch controller using the root locus approach. This may be due to the balanced gain selection for proportional and derivative gains to achieve a trade-off between rise time and overshoot. Hence, we decided to design a PD controller by manually tuning the gains. With the manual selection of controller gains, the pitch response of the octocopter for the desired pitch attitude command was obtained using MATLAB and has been included in the updated manuscript.

// 8. For the UAV and control systems, the IMU can strongly affect the performance. Provide a quantitative discussion of this aspect at least. //

Response to the comment:

In our attitude controller design, the gyro output from the flight controller board is taken as 0.86 volts/radian and this figure has been used in the attitude feedback loop. The gyro specifications considered in our study were based on the manufacturer’s data sheet. In real-time applications, to improve the performance of the control loops, the bandwidth of the gyro should be greater than the bandwidth of the attitude control loops, which was verified in our study.

// 10. Please add either stability and control derivatives or present all aerodynamic coefficients (i.e. Cx0, Cxu, Cxw, …) for a set of selected trim points (in a table). Was model stitching used? //

Response to the comment:

In our modeling, we do not use the state-space model, and for our simulation model, we considered the linearized equations. Hence, it was not required to introduce any stability derivatives in our study.

In our work, the model stitching method was not used for simulating the octocopter response. Moreover, we did not use a flight envelope for the simulation. In our simulation, we assumed the initial attitude to be zero before applying any inputs. We simulated the response of the octocopter in a MATLAB Simulink environment with inputs matching real-time user inputs.

// 11. The paper focuses on conceptual design and thus does not present results from real life application. This is fine, but it should be justified by presenting that this approach is used in other studies, e.g.: This approach is often used in flight mechanics as it reduces costs when new designs and system modifications are tested [R4 R5].

[R4] Topczewski S., Ĺ»ugaj M., Bibik P., „Impact of actuators backlash on the helicopter control during landing on the moving vessel deck”, Aircraft Engineering and Aerospace Technology, 2021

[R5] Lichota P., Noreña D. A.: "A Priori Model Inclusion in the Multisine Maneuver Design", 17th International Carpathian Control Conference, IEEE, Tatranska Lomnica, Slovakia, may 2016, pp. 440-445. //

Response to the comment:

Although the reviewer’s opinion could be a useful supportive description to our manuscript, we concluded that the recommended opinion and references do not perfectly fit to our manuscript scope.

Round 2

Reviewer 1 Report

The manuscript is greatly revised, but there is always room to improve. For example, do you really need to mention " The salt particles absorb fog and
smoke particles, creating an imbalance that allows for the dissipation of low visibility.
" in the abstract? In other words, is this something important in this study as a take-home message? In the end, authors claimed that " Thus, the proposed octocopter is structurally stable and can be efficiently used inall types of environmental applications." I would argue its operational capability under dust storm conditions. So would it be more appropriate to change 'all' to 'many'? These are just minor issues, but I suggest authors to spend time polishing it during the proof process.

Author Response

The manuscript is greatly revised, but there is always room to improve.

// 1. For example, do you really need to mention “The salt particles absorb fog and smoke particles, creating an imbalance that allows for the dissipation of low visibility." in the abstract? In other words, is this something important in this study as a take-home message? //

The statement was updated. Major two reasons proposed to clear the poor visibility issues are: 1. imposing of seawater, which is having high denser and relevant general properties, 2. with the help of propellers high rotodynamic effect and aerodynamic flow effect of C-D duct based storage tank, the turbulence formation and kinetic energy of the fluid particles, which are moves in and around the Octocopter are increased. Thus, the settlement of the foggy particles are pushed en route to the ground due to the presence of high energized turbulence [this provides the complete mixing and high pushing forces addition to the gravity force]. These observations were clearly stated in the updated statement.

// 2. In the end, authors claimed that “Thus, the proposed octocopter is structurally stable and can be efficiently used in all types of environmental applications." I would argue its operational capability under dust storm conditions. So would it be more appropriate to change 'all' to 'many'? These are just minor issues, but I suggest authors to spend time polishing it during the proof process. //

Yes. The comment from the reviewer was good and reasonable so it was accepted and corrected as per the suggestion of the reviewer.  Through FSI simulation, the ultimate stress induced inside the components of the Octocopter was computed, which is 0.40 MPa. The computed value was compared the ultimate load withstanding capability of the implemented lightweight materials and thereafter it was found out that the computed cum induced stress on Octocopter was highly lesser than the materials’ ultimate value. That was the reason for the above statements were previously written like that.   

Reviewer 3 Report

Thank you for providing responses to all my comments. 

Author Response

REVIEWER – 3

// 1. Does the introduction provide sufficient background and include all relevant references? Can be improved //

As per the requirement, the contents and references were updated.

// 2. Is the research design appropriate? Can be improved //

The design based contents and its novelties were updated.

// 3. Are the methods adequately described? Can be improved //

The methodology proposed section was briefly updated with some more technical contents.

// 4. Are the results clearly presented? Can be improved//

Yes Presented.

// 5. Are the conclusions supported by the results? Can be improved //

The conclusions were carefully checked and the contents are précised now.
